# Versican promotes T helper 17 cytotoxic inflammation and impedes oligodendrocyte precursor cell remyelination

Samira Ghorbani[1], Emily Jelinek[1], Rajiv Jain[1], Benjamin Buehner [1], Cenxiao Li[1], Brian M. Lozinski [1], Susobhan Sarkar[1], Deepak K. Kaushik[1], Yifei Dong [1], Thomas N. Wight[2], Soheila Karimi-Abdolrezaee [3], Geert J. Schenk[4], Eva M. Strijbis [5], Jeroen Geurts[4], Ping Zhang[6], Chang-Chun Ling[6] & V. Wee Yong [1]✉

Remyelination failure in multiple sclerosis (MS) contributes to progression of disability. The deficient repair results from neuroinflammation and deposition of inhibitors including chondroitin sulfate proteoglycans (CSPGs). Which CSPG member is repair-inhibitory or alters local inflammation to exacerbate injury is unknown. Here, we correlate high versican-V1 expression in MS lesions with deficient premyelinating oligodendrocytes, and highlight its selective upregulation amongst CSPG members in experimental autoimmune encephalomyelitis (EAE) lesions modeling MS. In culture, purified versican-V1 inhibits oligodendrocyte precursor cells (OPCs) and promotes T helper 17 (Th17) polarization. Versican-V1-exposed Th17 cells are particularly toxic to OPCs. In NG2[CreER]:MAPT[mGFP] mice illuminating newly formed GFP[+] oligodendrocytes/myelin, difluorosamine (peracetylated,4,4-difluoro-N-acetylglucosamine) treatment from peak EAE reduces lesional versican-V1 and Th17 frequency, while enhancing GFP[+] profiles. We suggest that lesion-elevated versican-V1 directly impedes OPCs while it indirectly inhibits remyelination through elevating local Th17 cytotoxic neuroinflammation. We propose CSPG-lowering drugs as potential dual pronged repair and immunomodulatory therapeutics for MS.

---

[1] Hotchkiss Brain Institute and Department of Clinical Neurosciences, University of Calgary, Calgary, Canada. [2] Benaroya Research Institute, Seattle, United States. [3] Department of Physiology and Pathophysiology, University of Manitoba, Winnipeg, Canada. [4] Department of Anatomy and Neurosciences, Amsterdam Neuroscience, MS Center Amsterdam, Amsterdam, The Netherlands. [5] Neurology, Amsterdam Neuroscience, MS Center Amsterdam, Amsterdam, The Netherlands. [6] Department of Chemistry, University of Calgary, Calgary, Canada. ✉email: vyong@ucalgary.ca

Multiple sclerosis (MS) is a chronic inflammatory disorder characterized by infiltration of T lymphocytes into the central nervous system (CNS) resulting in demyelination and axonal injury. Interferon (IFN)-γ-producing T helper (Th)1 and interleukin (IL)−17-expressing Th17 CD4[+] lymphocytes are considered to be key promoters of neuroinflammation and myelin damage, as are CD8[+] T cells and B lymphocytes[1–4]. Remyelination as a spontaneous repair response occurs extensively in many patients at early stages of MS but this process is not always efficient. Remyelination failure contributes to the axonal loss and progression of disability[5,6]. The failed repair process could be the consequence of ongoing toxic neuroinflammation and/or presence of inhibitors of oligodendrocyte lineage cells in lesions[6,7].

Various extracellular matrix (ECM) molecules including chondroitin sulfate proteoglycans (CSPGs) contribute to the inhibitory microenvironment of MS lesions[8–10]. An important subgroup of CSPGs is the lecticans that include 4 members: brevican, neurocan, aggrecan and versican. Versican itself has at least 4 isoforms. The lectican CSPGs are upregulated in MS lesions[11,12] and have been described to directly inhibit the differentiation of oligodendrocyte precursor cells (OPCs) into oligodendrocytes and to prevent remyelination[13–16]. The injection of the enzyme chondroitinase-ABC into the aged CNS promotes both OPC proliferation and differentiation[17]. In vivo inhibition of CSPG synthesis, deposition or signaling enhances remyelination post-injury[14,16,18,19]. However, which lectican CSPG member is particularly important in lesions is not clarified.

In addition to suppressing OPC differentiation, a commercially available CSPG mixture containing several lectican CSPGs enhances macrophage migration and production of pro-inflammatory cytokines such as tumor necrosis factor-α and interleukin-6[12,20,21]. Thus, CSPGs deposited into demyelinated lesions may provide the fuel that exacerbates inflammatory responses, but this remains to be demonstrated.

Amongst the lectican CSPGs, versican-V1 is profoundly elevated in lesions of lysolecithin-induced demyelination and at sites of infiltrating leukocytes in experimental autoimmune encephalomyelitis (EAE) and MS[12]. Not addressed is whether versican-V1 directly impairs OPCs, and whether it promotes neuroinflammation particularly Th17 immune responses capable of mediating CNS injury in MS[22,23]. Also unknown is whether impairing versican-V1 synthesis enhances remyelination in the EAE model, which is a crucial test as EAE has persisting innate and adaptive neuroinflammation typical of MS.

In the current study, we have sought to address critical gaps of knowledge on CSPGs: whether versican-V1 is associated in lesions of MS with reduced remyelination capacity, whether versican-V1 is directly inhibitory to OPCs, and whether CSPGs and versican-V1 locally regulate T cell plasticity particularly of Th17 cells that may then impair OPC function. T cells can be reprogrammed according to environmental cues at lesion sites[24,25] and it is not known if CSPGs could be such a lesional factor. Finally, we have employed the NG2[CreER]:MAPT[mGFP] mice that report on newly formed myelinating oligodendrocytes/myelin[26] to address whether the inhibition of CSPG production in EAE promotes remyelination. We tested Ac-4,4-diF-N-acetylglucosamine ("difluorosamine") that has high potency at reducing CSPG production in cultured cells[27]. Our results identify versican-V1 as a therapeutic target in inflammatory demyelinating lesions to reduce Th17 polarization and its toxicity to OPCs, and to promote extensive GFP[+] oligodendrocyte processes in the milieu of ongoing adaptive and innate neuroinflammation.

## Results

### MS lesions with high level of versican-V1 are associated with lower number of BCAS1[+] cells. Frozen brain autopsy sections

from three people with MS and one control individual were labeled with the versican-V1 antibody. Lesions were tracked by loss of luxol fast blue (LFB) myelin staining (Supplementary Fig. 1) and by prominent accumulation of CD45[+] immune cells. Two MS cases (MS-163, MS-352) had active lesions characterized by uniform distribution of immune cells throughout the lesions while the hypercellular edge of MS-230 indicated its chronic active nature. Versican-V1 immunoreactivity was markedly elevated in active and at the edge of chronic active lesions, closely associated with immune cells (Fig. 1a, b).

Next, brain tissue samples from 6 additional people with MS (Supplementary Table 1) comprising one active, four chronic active and four inactive lesions were immunohistochemically stained for versican-V1 and BCAS1 as a marker of premyelinating oligodendrocytes[28]. BCAS1[+] oligodendrocytes have been described as newly formed, premyelinating oligodendrocytes that are distinct from OPCs and mature oligodendrocytes, even though the majority of BCAS1[+] cells (76%) express a mature oligodendrocyte marker (CC1) and 16% of them are positive for an OPC marker (NG2)[28]. The first case had 2 demyelinated lesions in the same section (Fig. 1c) with markedly divergent amounts of versican-V1. The low versican-V1-containing lesion had higher number of BCAS1[+] cells than the high versican-V1-containing plaque that did not have detectable BCAS1[+] cells (Fig. 1c). Another case had 2 lesions where the versican-V1 expression was not as divergent. Here, the lower versican-V1-expressing lesion (diffuse stain, light brown, Fig. 1d upper panel) had more BCAS1[+] cells than the higher versican-V1 immunoreactive area (darker brown, Fig. 1d lower panel).

Next, versican-V1 level of lesion was qualitatively divided into low versus high expression based on relative difference to versican-V1 immunoreactivity in NAWM in the small section. The comparison across 6 MS autopsy cases with 19 regions of interest (fields of view) shows that the expression of versican-V1 is not dependent on the type of lesion, with high or low versican-V1 being spread across active, chronic active, or inactive white matter lesions (Fig. 1e). However, the number of BCAS1[+] remyelinating oligodendrocytes is inversely related to versican-V1 content, with high versican-V1 areas generally containing a lower number of BCAS1[+] cells (Fig. 1e).

We conducted spatial RNA sequencing (spRNAseq) (Jain et al., in preparation) of post-mortem MS brain tissue section which showed elevated levels of versican mRNA in the inactive demyelinated lesion, and to a lesser extent in the active core, compared to normal appearing white matter (NAWM) or to non-MS controls (Supplementary Fig. 2a, b). Expression level of BCAS1 appears inversely correlated with versican mRNA transcripts in MS lesions. The relatively low sensitivity of spRNAseq did not result in the detection of IL-17, RORγT or instructive T cell markers, and we were thus unable to describe relationship of versican to Th17 inflammation through this technique.

Previous work documenting that CSPGs are inhibitory for oligodendrocytes in culture have utilized either a mixed CSPG preparation or purified aggrecan[14,16]. Since versican-V1 is prominently elevated in MS lesions with low BCAS1[+] premyelinating oligodendrocytes, we plated murine (Fig. 2a) and human (Fig. 2f) OPCs onto versican-V1 in culture. Both mixed CSPG substrate (containing aggrecan, versicans, phosphacan and neurocan according to the manufacturer) and purified versican-V1 reduced the number of attached OPCs; of those adhered, process extension was reduced from the earliest point examined (6 h) (Fig. 2a–g).

While process outgrowth in culture is indicative of the potential of an oligodendrocyte to extend elaborate processes to contact and enwrap multiple axons during myelination in vivo,

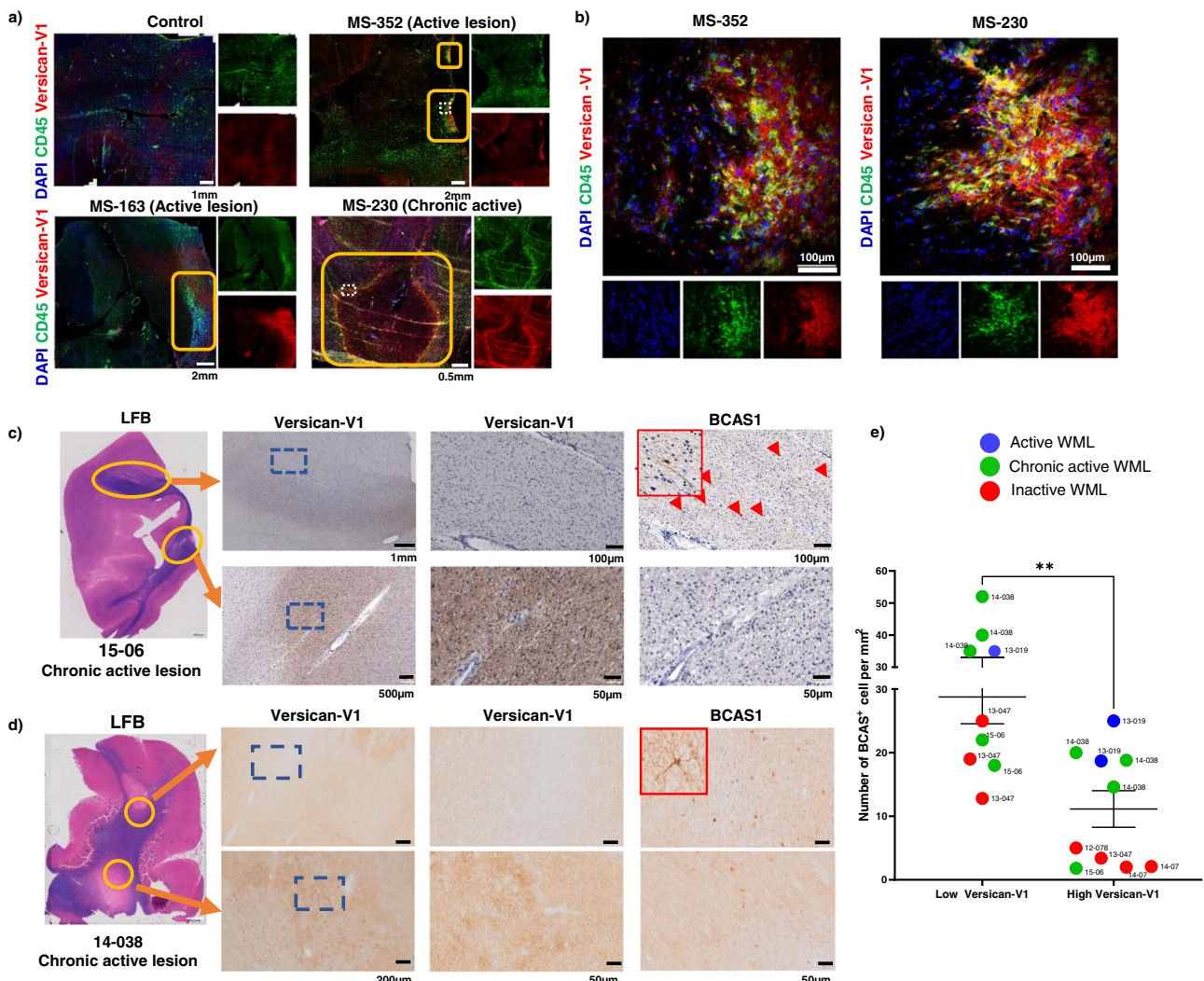

**Fig. 1 High level of versican-V1 in MS lesions is associated with low number of BCAS1+ cells. a**, **b** Immunofluorescent images comparing control human brain tissue (no documented neurological condition) and MS brain lesions from 3 cases (MS163, 230, 352) labeled with DAPI for cell nuclei (blue), CD45 for immune cells (green) and versican-V1 (red). Active lesion areas were tracked by prominent accumulation of CD45+ cells. Images were acquired with slide scanner **a** and laser confocal microscopy (b, z-stack). Yellow boxes indicate lesions and dotted white lines specify the areas shown at higher magnification in **b**. **c**, **d** Brain tissue samples from 6 people with MS described in Supplemental Table 1 were immunohistochemically stained for versican-V1 and BCAS1. Representative images from two different patients (15-06 and 14-038) are shown. White matter lesion (WML) areas were tracked by loss of luxol fast blue (LFB) staining. Yellow rings indicate the analyzed sites within the lesions and blue squares in each of the left panels (lower magnification) specify the areas shown at higher magnification to the right. Staining was repeated twice and showed the same results. **e** Bar graph comparing the number of BCAS1+ cells in 19 regions of interest (ROI) with high or low level of versican-V1 deposition, $n = 19$ ROIs from 9 lesions acquired from six individual MS brain samples. Data are represented as mean ± SEM. Two-tailed Mann–Whitney U-test; significance indicated as **$p = 0.0029$. Source data are in the source data file.

the maturation of OPCs into myelin basic protein (MBP)+ cells is another important feature of remyelination. Thus, we addressed the proportion of O4-expressing cells that matured into MBP+ oligodendrocytes at 72 h after plating, and this was decreased significantly on CSPGs or versican-V1 substrates (Fig. 2h–l).

Considering the effect of negatively charged CSPG surface on cell adhesion and subsequently on process outgrowth, positively charged poly-arginine peptide was used to counteract the negative charges of CSPGs. Although the neutralized substrate improved cell adhesion, the inhibitory effect of CSPGs on process outgrowth was not overcome (Supplementary Fig. 3b, c). Moreover, increasing the number of attached cells by high initial seeding density did not enhance the process growth on CSPGs, indicating suppressive role of CSPGs regardless of cell density (Supplementary Fig. 3d, e).

Collectively, the results highlight versican-V1 in MS lesions that is inversely correlated with the presence of BCAS1+ premyelinating oligodendrocytes. Moreover, in vitro, a versican-V1 substratum impairs OPC adhesion, morphological differentiation and their maturation into MBP+ oligodendrocytes.

**Elevation of versican-V1 and particular ECM proteins in EAE lesions**. The autopsy snapshot of MS cases did not permit a time course study. We thus examined the ECM in spinal cord white matter lesions of EAE mice (Fig. 3a, b) harvested at onset of clinical signs (day 12), peak (day 18) and post peak (day 40) of clinical severity. Elevation of versican-V1 was noted at the onset and peak of EAE followed by reduced but still detectable level at post peak EAE (Fig. 3c). Using Imaris 3D rendering, versican-V1

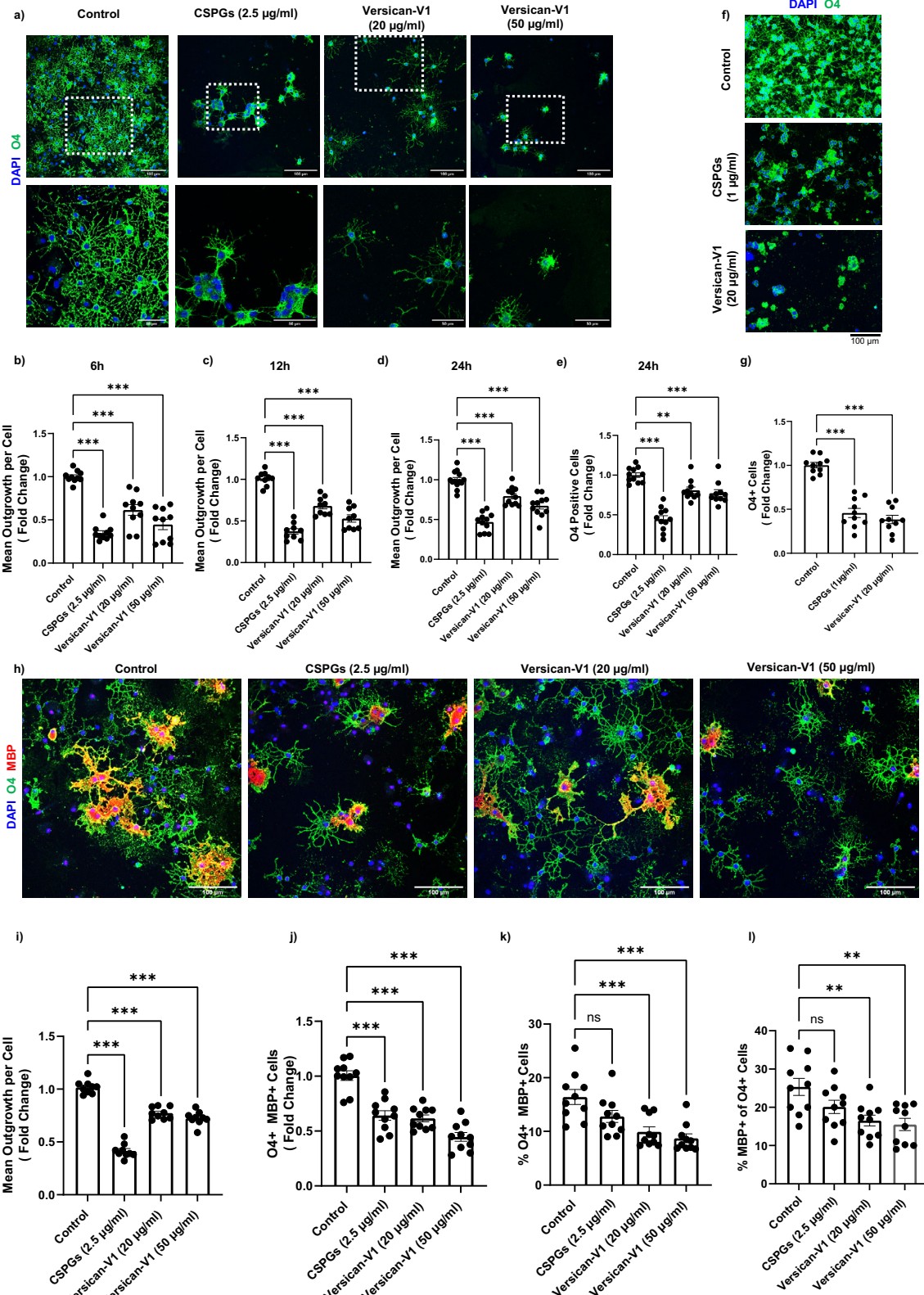

was found colocalized to CD45[+] immune cells although not all versican-V1-positive cells were CD45[+] (Fig. 3d). Moreover, versican-V1 was abundant in presumed extracellular spaces within the lesion and also associated within CD45[+] immune cells. Intracellular versican staining may reflect cellular production, as we previously found versican-V1 transcript in CD45[+] cells using in situ hybridization[12].

We evaluated tissue sections at the peak of EAE clinical disability (day 18) to address the relative elevation of versican-V1 and other ECM members (Fig. 3e–j). Sections were also stained for CD45 as a marker of immune cells to define the lesion area. The percent area occupied by ECM proteins within the CD45[+] lesions was compared to staining in normal appearing white matter (NAWM). We found increased levels of versican-V1,

**Fig. 2 OPCs are inhibited by CSPGs and versican-V1 in vitro. a** Representative images of mouse OPCs stained for the sulfatide O4 24 h after plating onto PBS (control), CSPGs and versican-V1 coated wells. Dotted white lines specify the areas shown at higher magnification in the lower panel. Scale bar, 100 μm (top panel) and 50 μm (bottom panel). **b–e** Bar graphs comparing the fold change of mean process outgrowth of mouse OPCs cultured on control, CSPG or versican-V1 after 6 h ($n = 9$ replicates), 12 h ($n = 9$) or 24 h ($n = 11$ replicates); and number of O4-expressing cells at 24 h ($n = 11$ replicates). Data are presented as mean ± SEM. Data acquired from 9 to 11 replicates over three separate experiments. For each well of cells, data from twelve images (field of views, FOVs) were averaged to a single data point. One-way ANOVA with Bonferroni post hoc; **$p < 0.01$, ***$p < 0.001$. **f, g** Representative images from human OPCs (scale bar, 100 μm) and bar graph showing the fold change of O4$^+$ cells from human OPCs cultured under different conditions. Data are presented as mean ± SEM, $n = 10$ replicates over three independent experiments, one-way ANOVA - Bonferroni post hoc; ***$p < 0.001$. **h** Representative images of mouse OPCs stained for O4 (green) and MBP (red) 72 h after plating onto PBS (control), CSPGs and versican-V1. Scale bar, 100 μm. **i–k** Bar graphs comparing the fold change of mean process outgrowth ($n = 9$ replicates), number and % of O4$^+$ MBP$^+$ cells ($n = 10$ replicates each), and **l** proportion of MBP-expressing cells among O4$^+$ cells ($n = 10$ replicates) of mouse OPCs cultured for 72 h. Data are presented as mean ± SEM. Data acquired from 9 to 10 replicates over three separate experiments. For each well, data from twelve FOVs were averaged to a single data point. One-way ANOVA - Bonferroni post hoc; **$p < 0.01$, ***$p < 0.001$. Source data are provided in the source data file.

fibronectin, thrombospondin and heparan sulfate proteoglycan in lesions while versican-V2 and aggrecan were unaltered (Fig. 3k–q). Together, these observations show that specific ECM members are deposited in the EAE lesions.

**CSPG mixture and purified versican-V1 shift T cell polarization toward Th17 cells in culture.** The presence of versican-V1 in MS and EAE lesions invites the assessment of whether it promotes pro-inflammatory T cell polarization that is a feature of MS[2,3]. We addressed this by isolating naïve CD4$^+$ T cells from mice and culturing them on ECM-coated plates (10 μg/ml) with cytokines/antibodies to generate different Th phenotypes (IFNγ$^+$ Th1, IL-17$^+$ Th17, FoxP3$^+$ Treg). Among Th1, Th17 and Treg subsets, the mixed CSPGs preferentially promoted Th17 polarization (Fig. 4a, b) and this was reproduced by 20 μg/ml purified versican-V1 (Fig. 4c).

We used the mixed CSPGs as a surrogate of versican-V1, given the limited supply of the latter. Compared to other ECM molecules elevated (fibronectin, thrombospondin, heparan sulfate proteoglycans) or trending towards an increase (fibrinogen) in EAE (see Fig. 3), only CSPGs elevated Th17 differentiation as determined by flow cytometry or ELISA (Fig. 4d, e). T cell activation, proliferation and viability showed no difference in CSPG-exposed T cells compared to control (Supplementary Figure 4a–f).

T cells can express different receptors interacting with CSPGs such as protein tyrosine phosphatase Sigma (PTPσ), leukocyte common antigen (LAR) and integrins[29]. Blocking the signaling pathway of PTP and LAR using intracellular sigma peptide (ISP) and intracellular LAR peptide (ILP)[18,19] could not reverse the CSPG increase of Th17 polarization. Indeed, blocking these signaling pathways elevated the frequency of Th17 cells (Fig. 4f) which is in agreement with a previous study[30]. We also used anti-integrin β1, β3, and β6 antibodies and determined that anti-integrin β3 neutralized the promoting effect of CSPGs on Th17 polarization (Fig. 4g).

Altogether, these results highlight the role of versican-V1 and CSPGs in Th17 differentiation, putatively through interaction with integrin β3 on T cells.

**An inhibitor of CSPG production, difluorosamine, reduces Th17 cells in the spinal cord of EAE mice.** We previously described difluorosamine to reduce the synthesis of CSPG in cultured cells; difluorosamine also lowered EAE severity in mice although levels of CSPGs in vivo were not determined[27]. To address if difluorosamine impacts Th17 representation in EAE, we initiated its daily treatment for five days from three days post onset of EAE clinical signs. Despite our best efforts at randomizing mice into two groups at the beginning of their

manifestation of clinical signs, mice appeared to be slightly divergent between the two groups although this was not statistically different (see Fig. 7 for another, longer term, experiment). This short-term treatment and prompt tissue harvest allowed us to study CNS T cell populations during peak clinical severity. Flow cytometry was then used to evaluate the composition of T cell subsets in the spinal cord and lymph nodes. The short period of treatment with difluorosamine reduced EAE severity marginally, which was associated with decreased CD4$^+$ T cell frequency in the spinal cord (Fig. 5a–c). Among the CD4$^+$ T cell population in the spinal cord, IL-17$^+$ cells showed a decrease in treated mice while IFN-γ$^+$ Th1 or FOXP3$^+$ regulatory T cells (Treg) remained unaffected. There was no significant change in IL-17-producing CD8$^+$ T cell (Fig. 5d). In contrast to the spinal cord, the frequency of different Th subsets in lymph nodes was not altered, ruling out a peripheral effect of difluorosamine on T cells that led to the spinal cord outcome. We excluded a direct effect of difluorosamine on Th17 polarization as noted in vitro (Supplementary Figure 4g).

These results link a potential inhibitor of injury-enhanced CSPG synthesis to locally decrease Th17 population in EAE, correspondent with less severe clinical disease.

**Versican-V1-exposed Th17 cells are more toxic to myelinating oligodendrocyte.** The increase of versican-V1 in MS and EAE lesions that locally elevates Th17 polarization raises the possibility that Th17 cellular activity could further be regulated by the CSPG microenvironment. Specifically, we addressed whether versican-V1-exposed Th17 cells could elevate OPC-killing activity to help account for the inverse relationship between premyelinating oligodendrocytes and versican-V1 in MS lesions (Fig. 1). 2D2 myelin oligodendrocyte glycoprotein (MOG)-reactive T cells were activated with MOG-loaded dendritic cells and differentiated to Th17 subset in the presence or absence of CSPGs (10 μg/ml) or versican-V1 (20 μg/ml) for 4 days. Th17 cells were then added to OPCs (Fig. 6a). Live imaging of calcein AM-labeled OPCs shows that few cells incorporate propidium iodide (PI), a small molecule that crosses the membrane of compromised cells[31], when exposed to soluble CSPGs (Fig. 6b). However, in the presence of Th17 cells, OPCs began to show PI incorporation from 2 h, signifying Th17 toxicity which became more apparent in the Th17 cells that were generated during CSPG or versican-V1 exposure (Fig. 6b, c). The proportion of PI-labeled OPCs was increased significantly following co-culture with CSPG- or versican-V1-exposed Th17 cells for 12 h (Fig. 6c).

In another experiment, the number of O4$^+$ or O4$^+$MBP$^+$ mature oligodendrocytes (Fig. 6d) incubated with Th17 cells, or Th17 generated during CSPG and versican-V1 exposure, were enumerated. Figure 6e shows that CSPG- or versican-V1-exposed Th17 cells reduced O4$^+$ and O4$^+$MBP$^+$ oligodendrocytes

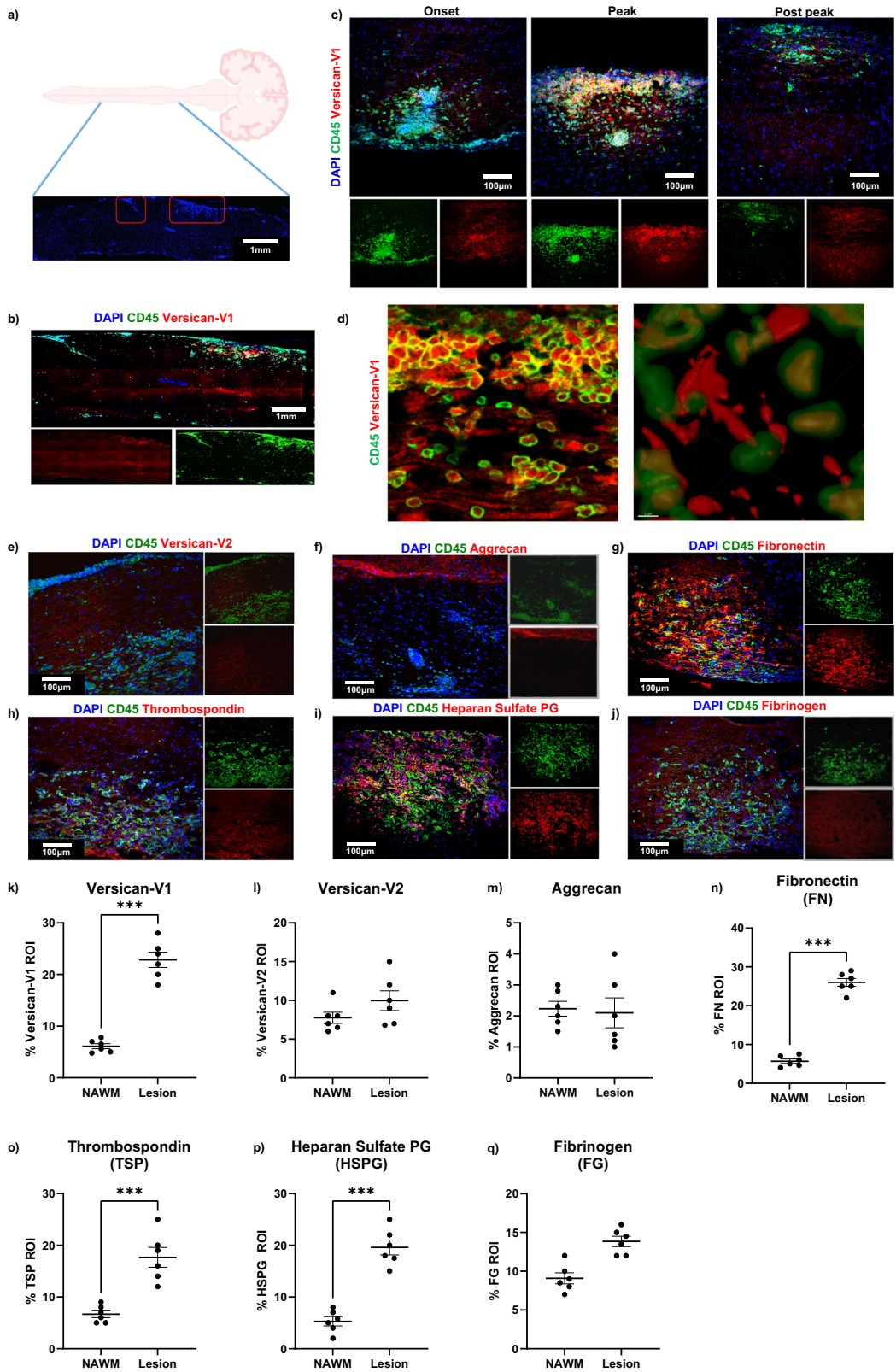

compared to Th17 cells alone. Process extension by oligodendrocytes was also markedly lowered by CSPG-Th17 cells (Fig. 6f), but not by Th17 lymphocytes generated during exposure to other ECM molecules.

We noted that the CSPG-enhanced Th17 toxicity was not non-specific to any T cell subset since CSPG-exposed Tregs did not overtly affect OPCs or oligodendrocytes (Supplementary

Fig. 5). As well, we ruled out an effect of CSPGs on the inflammatory profile of MOG-loaded dendritic cells (see schematic in Fig. 6a) by culturing both immature and mature dendritic cells in control or CSPG-coated plates. In this context, the frequency, maturation state and cytokine secretion of dendritic cells remained unaffected following CSPG versus control exposure (Supplementary Fig. 6).

**Fig. 3 Extracellular matrix proteins are upregulated in EAE lesions. a, b** Longitudinal sections of spinal cord from EAE mice were labeled with DAPI for cell nuclei (blue), CD45 (green) and ECM proteins (red). The lesion region of interest (ROI) was determined using CD45 (marker of immune cells) and a similar field used for normal appearing white matter (NAWM). Representative images of large area of spinal cord showing site of the lesions defined by hypercellularity (Created with BioRender.com) **a** and CD45$^+$cell accumulation **b** using Olympus VS110 Slide Scanner microscope. **c** Representative images of spinal cord sections from EAE mice at different phases of clinical disease severity including onset (day 12), peak (day 18) and post peak (day 40) stained for DAPI for cell nuclei (blue), CD45 for immune cells (green) and versican-V1 (red). **d** Using Imaris and 3D rendering, colocalization of versican-V1 and CD45$^+$ immune cells is observed although not all cells containing versican-V1 are CD45$^+$. We note that the two panels in **d** are not meant to be of adjacent sections or of same location, but rather to highlight the versican staining in the extracellular space and within CD45$^+$ cells. Experiments were repeated twice with similar results. **e–j** Expression levels of different ECM proteins in EAE lesions at day 18 in white matter of the spinal cord with quantitation shown in **k–q** when compared to NAWM. Data are presented as mean ± SEM. Data are acquired from 2 separate experiments with 3 mice per experiment ($n = 6$ mice total); each dot represents mean of 5 lesions analyzed per mouse, two-tailed unpaired Student's t test, Significance indicated as ***$p < 0.001$ for **k**, **n–p**. Source data are provided in the source data file.

Overall, these results highlight that versican-V1 can modify Th17 cells during their polarization, resulting in the Th17 cells being more harmful to oligodendrocyte lineage cells.

**CSPG-lowering drug improves presumed remyelination in EAE associated with decreased Th17 inflammation.** It has been difficult to study remyelination in EAE as the lesions appear at unpredictable location and because of the uncertainty of whether the oligodendrocytes within lesions are spared or remyelinating oligodendrocytes. This challenge has been overcome by the use of tamoxifen-inducible NG2$^{CreER}$:MAPT$^{mGFP}$ mice where membrane-associated GFP expression localizes to newly formed oligodendrocytes and myelin (Fig. 7a, b)[26].

EAE was induced in NG2$^{CreER}$:MAPT$^{mGFP}$ mice and spinal cords were isolated from tamoxifen injected control and EAE mice at peak (day 18) and post-peak of clinical severity (day 40); we refer to 'peak' as the period of pronounced clinical disability. To evaluate presumed remyelination during the course of EAE, immunofluorescence staining was performed on longitudinal tissue sections to allow evaluation of the entire thoracic spinal cord. All GFP$^+$ cells were immunoreactive for Olig2$^+$, a transcription factor specific to oligodendrocyte lineage cells (Fig. 7b). Few profiles of GFP-positivity were evident at day 18 sample that has high accumulation of CD45$^+$ cells but this increased in day 40 samples with reduced CD45$^+$ cells (Fig. 7c, d). GFP expression was not detected in EAE mice without tamoxifen injection indicating that the Cre recombinase was not leaky (Supplementary Fig. 7a).

To help with subsequent EAE analyses, we tested the NG2$^{CreER}$:MAPT$^{mGFP}$ mice further using demyelination induced by the toxin lysolecithin (LPC, lysophosphatidylcholine), which provides discrete phases of de- and remyelination[16,32] and therefore has been commonly used to study myelin repair. Indeed, in the NG2$^{CreER}$:MAPT$^{mGFP}$ mice with lysolecithin demyelination of the ventral white matter of the spinal cord, GFP expression was progressively increased from 7 to 21 days after demyelination, correlating with myelin repair (Supplementary Figure 7c–f). These findings corroborate the NG2$^{CreER}$:MAPT$^{mGFP}$ mice as a suitable model to study new oligodendrocytes and presumed remyelination in EAE.

Next, we tested difluorosamine in EAE. Difluorosamine was administered to NG2$^{CreER}$:MAPT$^{mGFP}$ mice daily starting from three days post onset of EAE clinical signs to exclude the effect of drug on immune cell infiltration. Mice were randomized into two groups according to their clinical scores to ensure similar starting clinical scores at the initiation of treatment. Figure 7e shows that difluorosamine progressively lowered disease severity compared to saline vehicle control.

At day 26, histological analyses of the spinal cord were performed. Given that treatment was started from peak clinical severity of EAE where substantial injury had already occurred, difluorosamine did not impact the extent of demyelination and axonal degeneration (Supplementary Figure 8a–c). Difluorosamine treated mice showed a non-significant increasing trend in CD45$^+$ immune cells (Fig. 7f, i), no change to the number of Iba1$^+$ microglia/macrophages and CD3$^+$ T cells (Fig. 7g, j; Supplementary Figure 8d, e), while the number of CD4$^+$ T cells was reduced (Fig. 7h, k).

Considering the increasing trend in number of CD45$^+$ immune cells, we resorted to PCR analyses to compare selected phenotypic markers of microglia/macrophages and T cells. The expression of pro-inflammatory (IL-17, IL-1α) and pro-oxidative (iNOS) transcripts in lumbar cord was significantly reduced in difluorosamine-treated EAE mice whereas IL-10 mRNA level was elevated (Fig. 7l–s). These results indicate a shift in the balance of inflammatory and regulatory cells in difluorosamine treated mice.

We found that difluorosamine-treatment resulted in substantially higher area of the entire thoracic spinal cord white matter occupied by GFP$^+$ myelinating oligodendrocytes and presumed new myelin sheaths (Fig. 8a–c). When we confined analysis to the well-delineated EAE lesions in thoracic cord, the increase in GFP$^+$ processes by difluorosamine treatment was also noted (Fig. 8d). GFP$^+$ oligodendrocytes in difluorosamine-treated mice had features of actively remyelinating cells with several processes suggestive of alignment along axons, while many GFP$^+$ cells in control mice displayed short and irregularly arranged processes (Fig. 8e, f). In addition to increased number of oligodendrocyte lineage cells (Olig2$^+$) and mature oligodendrocyte (olig2$^+$CC1$^+$), difluorosamine-treatment resulted in higher number of GFP$^+$ mature oligodendrocytes within the lesion (Fig. 8g–j)

Moreover, in these lesional regions of interest, the immunoreactivity for total CSPGs and also versican-V1 was lowered by difluorosamine (Fig. 8k–m). Indeed, there was a significant negative correlation between GFP expression and CSPG imunoreactivity in lesional regions of interest across all mice (Fig. 8n). A slight effect of difluorosamine on reducing the levels of HSPGs has been shown previously by our group[27]; however, no change was found in the present study (Supplementary Fig. 8f, g).

These results highlight that by lowering CSPG content within lesions, difluorosamine reduced Th17 neuroinflammation and promoted GFP$^+$ profiles indicative of remyelination in EAE.

**Discussion**

MS is a chronic and progressive inflammatory neurological disorder in which myelin sheaths and axons in the CNS are damaged by several types of immune cells including Th17 cells[4]. Findings from EAE studies have highlighted that Th17 cells can be reprogrammed at the site of inflammation[1,24,25]. However, local environmental cues and molecular mechanisms underlying T cell plasticity and perturbed T cell polarization at lesions have yet to be elucidated. Here, we provide evidence that the lectican CSPGs deposited in MS lesions[11,12], and specifically versican-V1, have

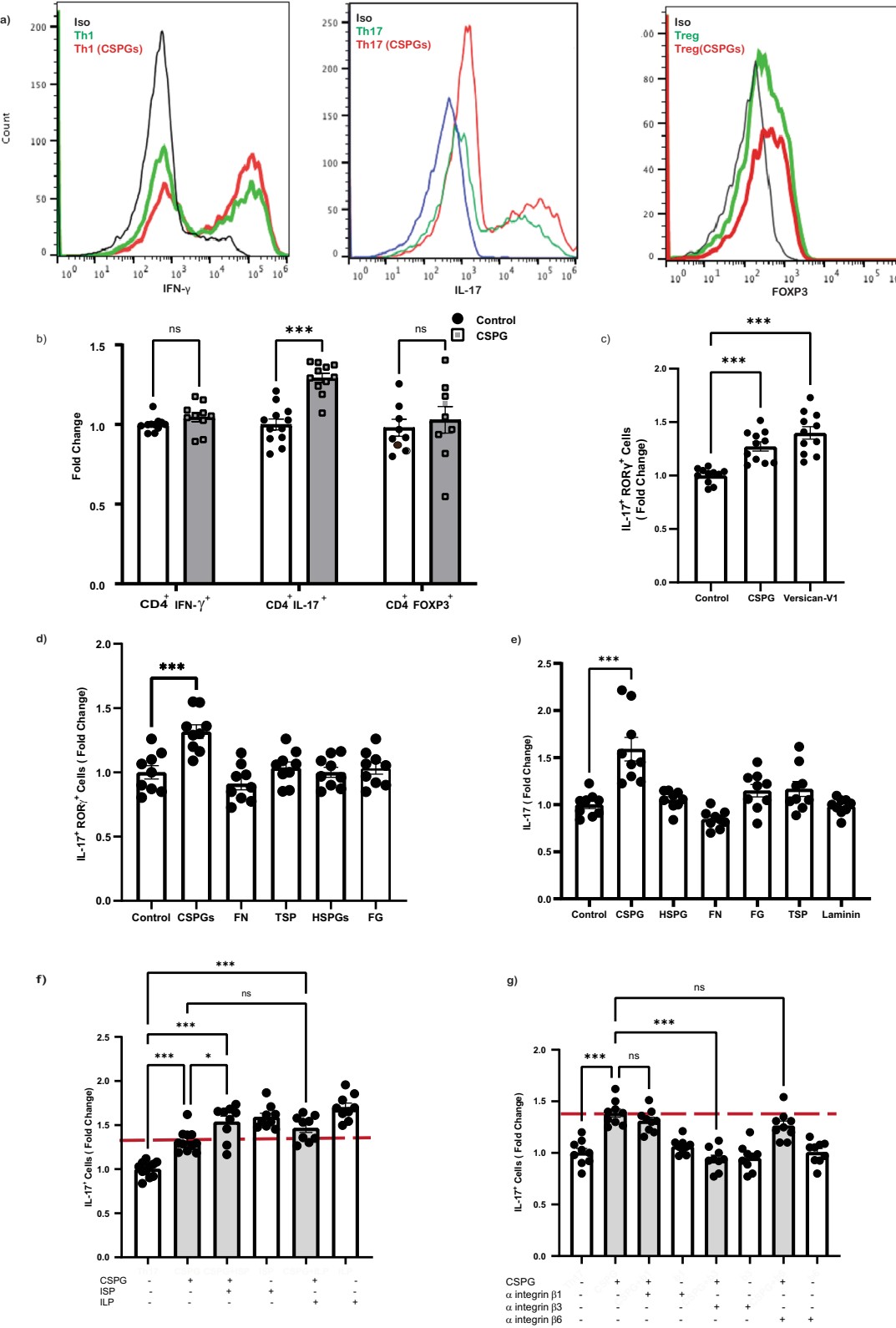

this capacity as shown by increased Th17 generation in culture when CD4$^+$ T cells are polarized in the presence of a mixed CSPG preparation and purified versican-V1. These findings are in accordance with a previous study, in which the chondroitin sulfate-A chain treatment before the onset of clinical signs exacerbated EAE severity accompanied by elevation of IL-17 in splenocytes[33]. However, the direct effect of CSPGs on Th17

differentiation and whether it could change the balance of T cell subsets in the CNS remained unclear. We have now addressed these questions herein. When we employed difluorosamine that we have previously shown to reduce the cellular production of CSPGs in culture[27], and which we corroborate herein to diminish versican-V1 in the spinal cord of EAE mice (Fig. 8), the level of IL-17 expressing CD4$^+$ cells was lowered locally in the spinal

**Fig. 4 CSPGs and its prominent member, versican-V1, promote Th17 differentiation and IL-17 production.** To assess the effect of ECM molecules on T cells, naïve CD4[+] T cells were cultured on ECM-coated wells (10 µg/ml) and were then activated and polarized to Th1, Th17 or Treg subset. The frequency of IFN-γ, IL-17 or FOXP3 expressing T cells was determined using flow cytometry 4 days later. **a** Representative histograms showing the frequency of IFN-γ, IL-17 or FOXP3 immunopositive T cells in control (green) or following CSPG treatment (red); isotype antibody control (blue) is also displayed. **b** The fold change of Th1 (71.7% Vs & 72.13%; $n = 10$ replicates), Th17 (18.64% Vs 23.61%; $n = 11$ replicates) and Treg (21.34% Vs 21.98%; $n = 9$ replicates) frequency is shown as a bar graph across 3 experiments. Data are presented as mean ± SEM, two-way ANOVA - Bonferroni post hoc; ***$p < 0.001$. **c, d** Bar graph presenting the percentage of IL-17[+]RORγ[+] Th cells resulting from a commercial mixed CSPG preparation (10 µg/ml) compared to purified versican-V1 (20 µg/ml) (Control:9.6%, CSPG:12.31%, Versican-V1: 13.53%) **c** or other ECM molecules (10 µg/ml) **d**; $n = 9$ replicates over three separate experiments. **e** Level of IL-17 in the conditioned medium of Th17 polarized cells measured by ELISA (Control: 1948 pg/ml, CSPG: 2800 pg/ml); $n = 9$ replicates over three separate experiments. **f** Frequency of Th17 cells after exposing cultures to signaling pathway inhibitors of CSPG receptors (PTPσ and LAR) using intercellular peptide (ISP and ILP, respectively, 2.5 µM), $n = 11$ replicates for control and CSPGs, $n = 9$ replicates for other conditions, over three separate experiments. **g** Frequency of Th17 cells after exposing cultures to function blocking antibodies to the integrin β1, β3, or β6 (50 µg/ml); $n = 9$ replicates over three separate experiments. Data presented as mean ± SEM. One-way ANOVA - Bonferroni post hoc; *$p = 0.01$, ***$p < 0.001$. Source data are provided in the source data file.

cord (Fig. 5). That difluorosamine treatment from peak clinical severity of EAE did not affect Th17 cells in the lymph node suggests the importance of the local spinal cord microenvironment in dictating the polarization of immune cells.

While remyelination occurs in MS, the repair process is not efficient likely due to ongoing inflammation or inhibitors in lesions that prevent the recruitment and/or differentiation of OPCs[6,34]. Overcoming inhibitors present within lesions has been proposed as a potential therapeutic approach for MS[7]. We have implicated CSPGs as inhibitors of the differentiation of OPCs and remyelination[14,16], but the crucial role of versican-V1 on OPCs could not be addressed because of the previous unavailability of purified versican-V1 protein. We have rectified this herein, and demonstrate that purified versican-V1 in culture inhibits the adhesion, morphological differentiation and myelin production of OPCs. In MS lesions, high versican-V1 expressing regions do not have abundance of BCAS1[+] cells which tend to reside in areas with low V1 content (Fig. 1). BCAS1 appears transiently in oligodendrocytes just before they myelinate axons[28]. Our data are in line with a previous study showing lower levels of versican in the gray matter compared to the white matter region of leukocortical MS lesions, which was correlated with better remyelination in the gray matter region[35]; the versican isoform was not defined in that study.

We note that while the mixed CSPG preparation is very inhibitory for OPC process outgrowth at 2.5 µg/ml, much higher concentration of purified versican-V1 (20 µg/ml) is required (Fig. 2). A possible reason for this is that the mixed CSPGs contain a variety of lectican CSPGs (aggrecan, brevican, neurocan and versican, according to the manufacturer) which may act in concert to inhibit OPCs. Moreover, aggrecan has extensive glycosaminoglycan chains, far exceeding that of versican, and the additional negative charges should more effectively impair adhesion since the attachment of OPCs onto CSPGs is neutralized by the positively charged poly-arginine peptide (Supplementary Fig. 3b, c).

Various T cell subsets impact remyelination differently. Regulatory T cells (Tregs) enhance remyelination whereas Th17 cells prevent myelin reformation[22,36,37]. In vivo transfer of myelin-reactive Th17 cells reduces endogenous remyelination in a toxin-induced demyelination mouse model[22]. A recent study shows the direct contact between CNS-infiltrating Th17 cells and oligodendrocytes in EAE and MS lesions results in oligodendrocyte death and impaired remyelination through release of glutamate[38]. IL-17 from Th17 cells induces NOTCH1 (Notch homolog 1, translocation-associated) signaling in OPCs, resulting in apoptosis and reduced differentiation[39]. We corroborate the toxicity of Th17 cells on OPCs, and further demonstrate that the exposure of Th17 cells to versican-V1 during their differentiation elevates

their capacity to kill OPCs. Thus, the lesion content of versican-V1 not only affect Th17 polarization and OPC differentiation, but they endow the Th17 cells with greater capacity to kill OPCs. These results emphasize the importance of overcoming versican-V1 in lesions. The mechanisms underlying increased toxicity of versican-V1-exposed Th17 remain to be investigated in future studies.

Several means to reduce lesional CSPGs have been employed, and a common one is the local injection of the enzyme chondroitinase-ABC to remove the glycosaminoglycan chains of CSPGs[40,41]. However, this leaves the core protein intact, with uncertain impact on immune and oligodendroglial cells, and the local injection would not be applicable to a multifocal lesion condition such as MS. We screened several glucosamine analogs and described difluorosamine as the most potent in reducing CSPG production by cells in culture[27]. To test the overall therapeutic effects of difluorosamine, including on myelin formation, the EAE model with attendant adaptive and innate immunity, and histological hallmarks of MS would be important. Although the demonstration of thinner myelin sheaths by electron microscopy has become the gold standard for identifying sites of remyelination in localized toxin-induced demyelination models, it does not provide a ready means to study remyelination in EAE due to the uncertain location of lesions; thus, the microscale area of analysis in electron microscopy could easily miss a repairing lesion or under-report on remyelination across the spinal cord[6,42]. Also, studies on myelin reformation in EAE have been hampered by the difficulty of assessing whether oligodendrocytes and myelin found in a lesion are spared or regenerated. The availability of the NG2[CreER]:MAPT[mGFP] mice[26] has resolved these challenges, as newly formed oligodendrocytes from NG2 precursors prominently elevate microtubule associated protein tau (MAPT) which is required for process outgrowth and the transport of myelin basic protein mRNA to distal processes to form myelin[43]. Thus, the GFP signal driven by the tau promoter in oligodendrocytes informs on newly formed oligodendrocytes and nascent myelin[26], as we validated in the lysolecithin lesion with defined de- and remyelination phases (Fig. S7).

Using the tamoxifen-treated NG2[CreER]:MAPT[mGFP] mice induced for EAE, we found that mice administered difluorosamine had reduced EAE severity correspondent with lower versican-V1 content and frequency of Th17 cells in spinal cords, and more newly formed oligodendrocytes with elaborate processes (Figs. 7 and 8). While the GFP[+] signal is associated with nascent myelin[26], we did not conduct electron microscopy analysis to definitively establish remyelination in the fluorosamine-treated mice. Thus, we have referred to the extensive GFP[+] profiles as 'presumed' remyelination.

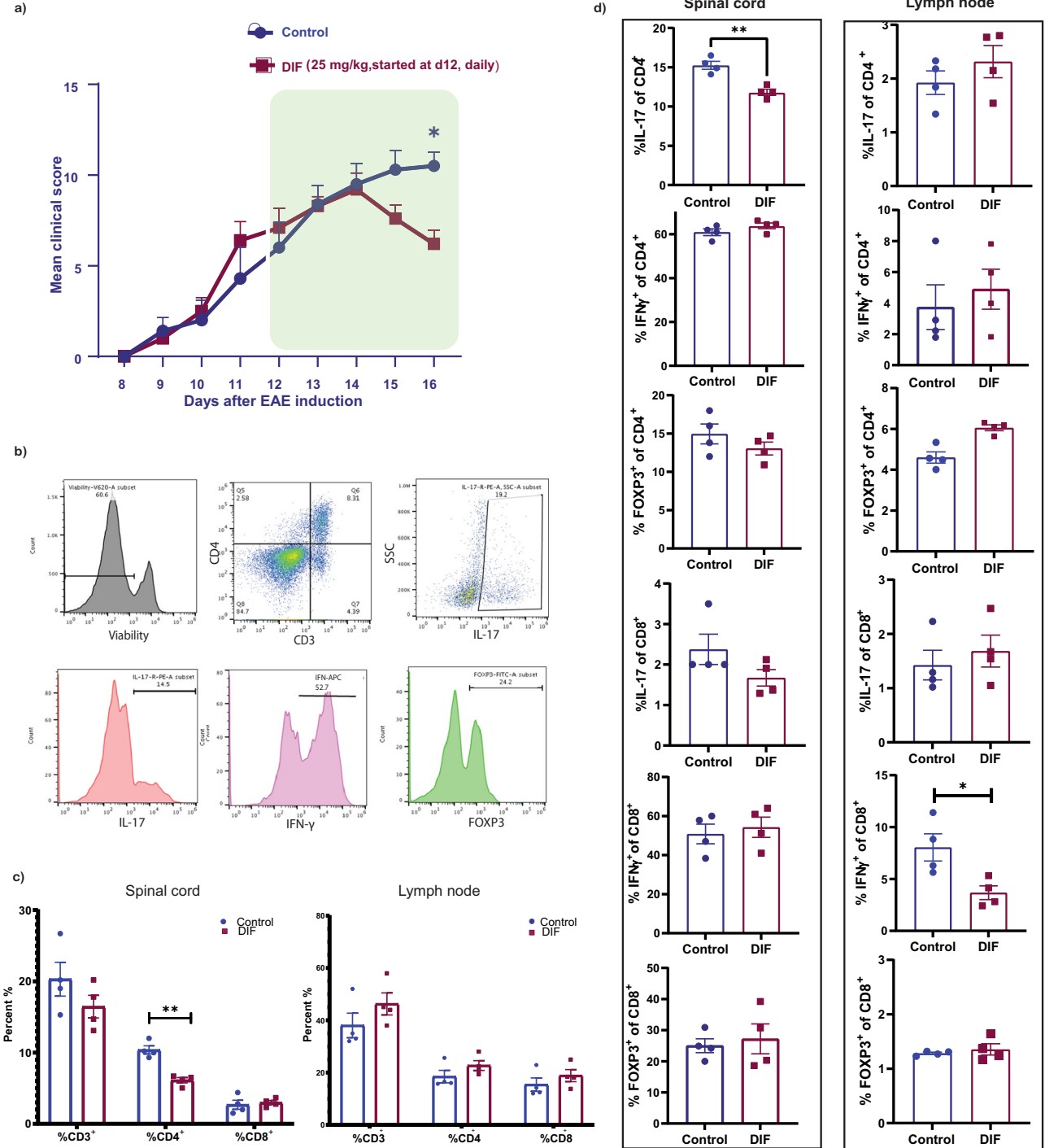

**Fig. 5 Difluorosamine reduces Th17 population in the CNS during EAE. a** A CSPG-lowering drug, difluorosamine (DIF), was administered at 25 mg/kg to mice daily starting three days post onset of EAE for 5 days. Data are presented as mean ± SEM of $n = 10$ mice per group pooled from 2 independent experiments. Results were analyzed using two-way repeated-measures ANOVA with Sidak's post-hoc test; *$p = 0.02$. Average EAE clinical score of mice is shown with treatment period indicated by green box. **b** The frequency of different T cell subsets was determined using flow cytometry. **c** Bar graphs showing the frequency of CD3$^+$, CD4$^+$ and CD8$^+$ cells; and **d** IFN-γ$^+$, IL-17$^+$ or FOXP3$^+$ T cells within the CD4$^+$ or CD8$^+$ populations in the spinal cord and lymph nodes of DIF or vehicle treated mice. Data are presented as mean ± SEM. $n = 4$ mice per group for **c**, **d**, and results were reproduced in a second experiment (source data file). Two-tailed unpaired Student's t test; **$p = 0.003$ for **c**, and *$p = 0.025$ or **$0.0017$ for **d**. Source data are provided in the source data file.

The association between T cell subsets and levels of versican-V1 within lesions may be studied further using RNAscope that provides better sensitivity and specificity (relative to spatial RNA sequencing) to measure low-abundance RNA biomarkers such as IL-17 and RORγT. In addition to T cells, the changes in innate immune responses and how they impact remyelination have yet to be explored thoroughly in the context of versican biology. Our future studies using versican conditional knock out mice would help to clarify the specific effect of versican rather than total CSPGs on immune cells during remyelination.

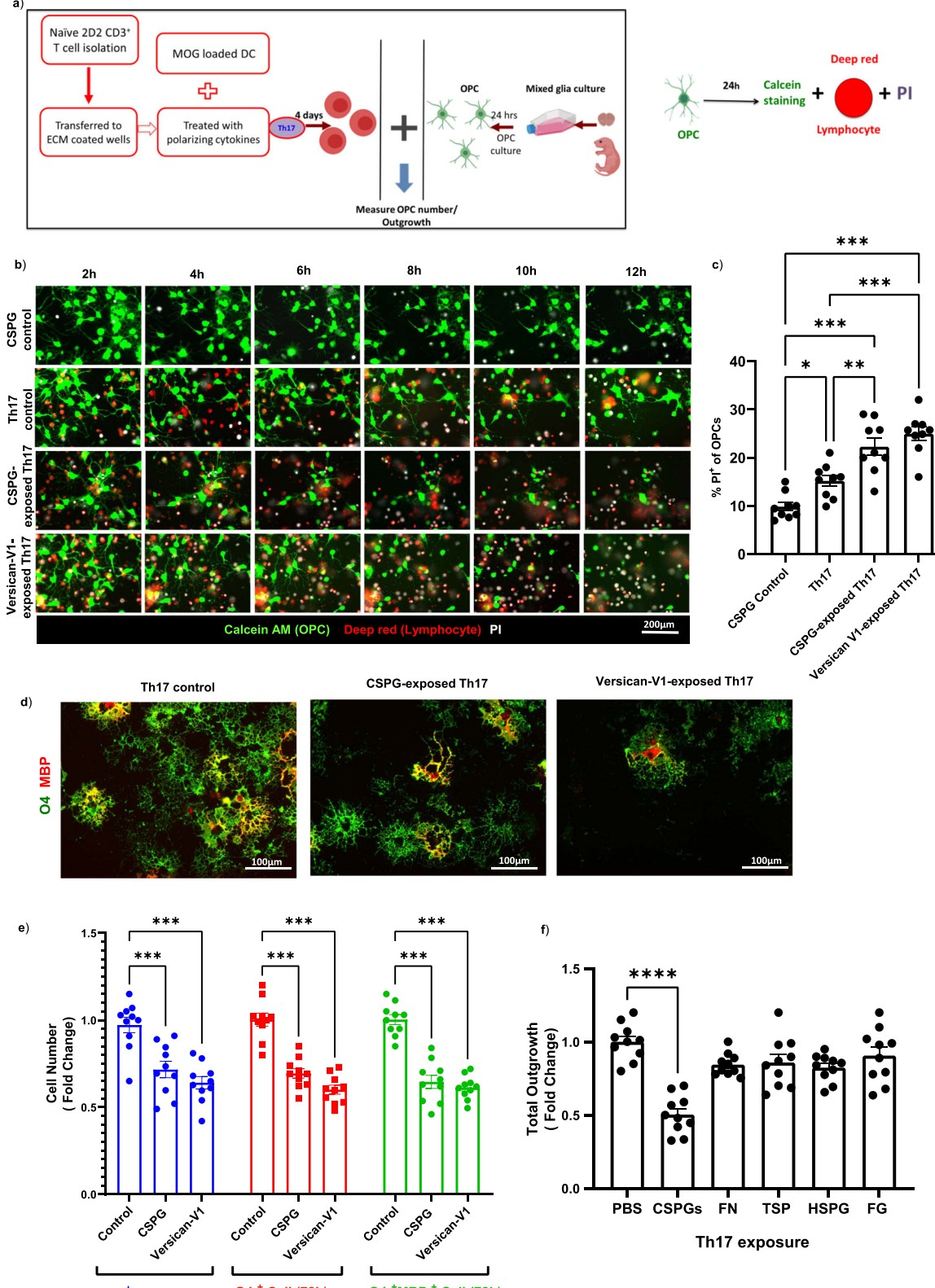

We note that while versican-V1 appears to be the predominant lectican CSPG in EAE lesions, we did not evaluate if it is the sole or predominant CSPG in MS lesions. Other lectican CSPGs may play similar roles in MS as versican-V1, and this remains to be investigated in future studies.

In conclusion, we provide evidence for the role of CSPGs and versican-V1 in Th17 cell polarization including in EAE lesions.

We found that mixed CSPGs and versican-V1 have multiple actions that would retard remyelination in MS, including directly inhibiting OPC differentiation, augmenting Th17 neuroinflammation, and elevating the propensity of Th17 cells to kill OPCs. Thus, the CSPG-reducing drug, difluorosamine, ameliorated clinical severity in EAE associated with lower Th17 neuroinflammation while enhancing oligodendrogenesis and

**Fig. 6 CSPG-exposed Th17 cells are more toxic to oligodendrocytes. a** 2D2 T cells were isolated and activated with MOG-loaded dendritic cells and differentiated to Th17 subset in ECM-coated plates and they were then added to OPCs for 24 h. Schematic diagram of experiment procedure is shown (Created with BioRender.com). **b, c** Live imaging of labeled OPC-T cell co-culture was performed with PI in the culture medium, and images at different time points are displayed. **b** Representative images of OPCs (green); PBS-, CSPG- or versican-V1-exposed Th17 cells (red); and cellular PI (white) are provided. To rule out a direct effect of CSPG in mediating death of OPCs, CSPGs (10 μg/ml) was also added to the culture medium of OPCs (CSPG control). **c** Bar graph showing the proportion of PI labeled OPCs after 12 h, $n = 9$ replicates over three separate experiments. Data are presented as mean ± SEM. One-way ANOVA - Bonferroni post hoc; significance indicated as *$p = 0.03$, **$p = 0.005$, ***$p < 0.001$. **d, e** In a separate co-culture experiment, cells were not labeled with cell trackers and instead were stained for O4 (green) and MBP (red) markers after 24 h. OPC cell number or process outgrowth were measured using ImageXpress. **d** Representative images from OPC co-culture with PBS (Control)-, CSPG- or versican-V1-exposed Th17 cells and **e** bar graph comparing the fold change of O4$^+$ OPCs and mature MBP$^+$ oligodendrocytes between PBS-, CSPG- or versican-V1-exposed Th17 cells after 24 or 72 h; $n = 10$ replicates over three separate experiments. Data are presented as mean ± SEM. Two-way ANOVA - Bonferroni post hoc; significance indicated as ***$p < 0.001$. **f** Quantification of OPC process outgrowth following co-culture with Th17 cells exposed to different ECM proteins, $n = 10$ replicates over three separate experiments. Data are presented as mean ± SEM. One-way ANOVA- Bonferroni post hoc; significance indicated as ****$p < 0.001$. For each replicate, data from twelve images (FOVs) were averaged to a single data point. Source data are provided in the source data file.

presumed remyelination. We propose CSPG-lowering drugs as potential dual pronged repair therapeutics that directly affect OPCs, and that indirectly antagonize Th17 roles in neuroinflammation and oligodendrocyte injury.

## Methods
All experiments were performed with ethics approval (protocol number AC21-0174) from the Animal Care Committee at the University of Calgary under regulations of the Canadian Council of Animal Care.

**MS specimens**. Postmortem frozen brain tissues from people with MS and healthy control brain tissue were obtained from The Multiple Sclerosis and Parkinson's Tissue Bank situated at Imperial College, London (https://www.imperial.ac.uk/medicine/multiple-sclerosis-and-parkinsons-tissue-bank). This bank has been approved as a Research Tissue Bank by the Wales Research Ethics Committee (Ref. No. 18/WA/0238). Secondary progressive MS tissues used for spRNAseq were obtained from Dr. Alex Prat, University of Montreal with full ethical approval (BH07.001, Nagano 20.332–YP) and informed consent as approved by the CRCHUM and University of Montreal research ethics committee. Paraffin-embedded sections from autopsied MS subjects were from the Netherlands Brain Bank (https://www.brainbank.nl/), Amsterdam. All samples at their local sites were collected with full informed consent for autopsy, and their use for research has been approved by local institutional ethics committee. The use of these human tissues in Calgary for research was approved by the Conjoint Health Research Ethics Board at the University of Calgary (Ethics ID REB15-0444).

MS sections were characterized using Luxol fast blue (LFB) and Hematoxylin & Eosin (H&E) to identify lesions. In addition, a combination of CD45, MBP, and pan-neurofilament (NF) staining was used to determine the activity of lesions. The Amsterdam cases (Fig. 1c–e) were prescreened in Holland for the presence or absence of BCAS1. To ascertain whether a given lesion has high or low versican immunoreactivity, we have compared the lesional staining intensity to the intensity of the normal appearing white matter within that tissue section. This avoids differences in staining intensity across slides, or the background stain in one slide versus another.

**Mice**. Female C57Bl/6 J mice (6-8 weeks old) and litters from pregnant CD1 mice (P1-P2) were purchased from Charles River and used for in vitro leukocyte or OPC cultures, respectively. NG2$^{CreER}$ (JAX 008538) mice and Tau$^{mGFP}$ (JAX 021162) mice aged 6 to 8 weeks were acquired from Jackson Laboratories and bred in University of Calgary Animal facility to produce female NG2$^{CreER}$:MAPT$^{mGFP}$ mice; the 2D2 TCR (TCR$^{MOG}$) transgenic mice (JAX 006912) were also from Jackson Laboratories. Mice were housed between 21 and 23 degrees Celsius, in low humidity, with 12 h light and 12 h dark cycle from 7 am light and starting 7 pm dark.

For EAE experiments or lysolecithin demyelination, the NG2$^{CreER}$:MAPT$^{mGFP}$ mice at 8–10 weeks were injected intraperitonially with 2 mg tamoxifen (100 μL of 20 mg/mL tamoxifen in corn oil) (Sigma) daily for 5 consecutive days to induce recombination. EAE or demyelination was then induced in mice at least 5 days after the last tamoxifen injection to allow tamoxifen to be washed out. Genotyping was conducted using protocols provided online through Jackson Laboratories.

**Experimental autoimmune encephalomyelitis (EAE) induction**. Female C57BL/6 wildtype mice (Jackson Laboratories) or NG2$^{creER}$:MAPT$^{mGFP}$ mice (8–12 weeks old) were injected subcutaneously with 200 μg MOG35-55 peptide (synthesized by Protein and Nucleic acid facility, Stanford University) emulsified in complete Freund's adjuvant (CFA) (Thermo Fisher Scientific) containing 10 mg/ml of heat inactivated *Mycobacterium tuberculosis* H37RA (Sigma-Aldrich); injections were at

one site into each hind flank. On the day of immunization and 48 h later, animals received intraperitoneal injections of 300 ng of pertussis toxin (List Biological Laboratories).

Clinical signs of EAE were evaluated daily using a 0 to 15 point scoring scale[44]. For the tail, 0 signifies no signs, a score of 1 represents a half-paralyzed tail and a score of 2 reflects a fully paralyzed tail. For each of the hind- or forelimbs, 0 is for no signs, a score of 1 is given to a mouse with a weak or altered gait, 2 represents paresis, while a score of 3 reflects a fully paralyzed limb. Mortality corresponds to a score of 15. Spinal cord tissues were dissected from EAE mice at three time points following MOG induction: onset (first day of the appearance of clinical signs, Day 12 post-immunization), at the peak of clinical severity (Day 18 post-immunization), and a late phase referred here as post-peak (Day 40 post-immunization). Mice were euthanized with ketamine (100 mg/kg) and xylazine (10 mg/kg) injected intraperitoneally and then perfused with PBS through the left ventricle of the heart. Following PBS-perfusion, lumbar and thoracic spinal cord were dissected for immunohistochemistry and RNA extraction, respectively. For FACS experiments, the whole spinal cord was used.

For diflurosamine treatment, mice were randomized into two groups of ten mice on day of MOG$_{35-55}$ immunization. Intraperitoneal treatment with either diflourosamine (25 mg/kg, dissolved in saline) or vehicle (saline) was started at the peak of disability and continued once a day until sacrifice.

All drug administration and EAE clinical scoring and analyses were done blinded to treatment. All procedures were performed according to the Canadian Council of Animal Care guidelines.

**Lysolecithin-induced demyelination**. Lysolecithin was injected into the ventral spinal cord to induce experimental demyelination[16]. Mice were first anesthetized with ketamine and xylazine (100 mg/kg and 10 mg/kg, respectively) administered intraperitoneally. The analgesic buprenorphine (0.05 mg/kg) was injected subcutaneously prior to surgery and 12 h post-surgery. A 3 cm incision was made between the shoulder blades, and fat and muscle were separated using retractors. Connective tissue between the T3-T4 vertebrae were blunt dissected to expose the dorsal spinal cord and the meninges were removed using a 30-gauge metal needle. To induce focal demyelination in the ventrolateral white matter of the spinal cord, 0.5 mL of 1% lysolecithin (LPC) (Sigma L1381) was injected at a rate of 0.25 μl/min over 2 min. A 10 μl Hamilton 34-gauge needle was inserted 1.3 mm into spinal cord to induce the injury in the ventrolateral side. Mice were then sutured and monitored until recovery. Mice were sacrificed at 7,14-, and 21-days post-injection of lysolecithin with a lethal dose of ketamine/xylazine. Animals were then perfused with 10 mL of PBS and 10 mL of 4% paraformaldehyde (PFA). The lower cervical and upper thoracic section of the spinal cord was dissected.

**Spinal cord tissue harvest**. For RNA extraction, spinal cords were immediately flash frozen in liquid nitrogen and stored at -80 °C. For immunohistochemistry, spinal cords were post-fixed overnight in 4% PFA or Periodate-Lysine-Paraformaldehyde PLP at 4 °C, then transferred into 30% sucrose solution for 72 h. Tissues were then frozen in FSC 22 Frozen Section Media (Leica). Using a cryostat (ThermoFisher Scientific), spinal cord blocks were cut coronally or longitudinally into 20 μm sections, collected on to Superfrost Plus microscope slides (VWR) and stored at -20 °C prior to staining.

**Immunofluorescence staining**. Slides were thawed at room temperature for thirty min, then hydrated with PBS for five min. Blocking of tissue sections was performed using horse serum blocking solution (0.01 M PBS, 10% horse serum, 1% bovine serum albumin (BSA), 0.1% cold fish skin gelatin, 0.1% Triton-X100, and 0.05% Tween-20) for 1 h at room temperature. Slides were then incubated with diluted primary antibodies overnight at 4 °C. Next, slides were washed three times, 5 min each with PBS containing 0.25% Tween-20 and then incubated with fluor-ophore conjugated secondary antibodies (1:400) and 1 μg/ml of DAPI suspended in

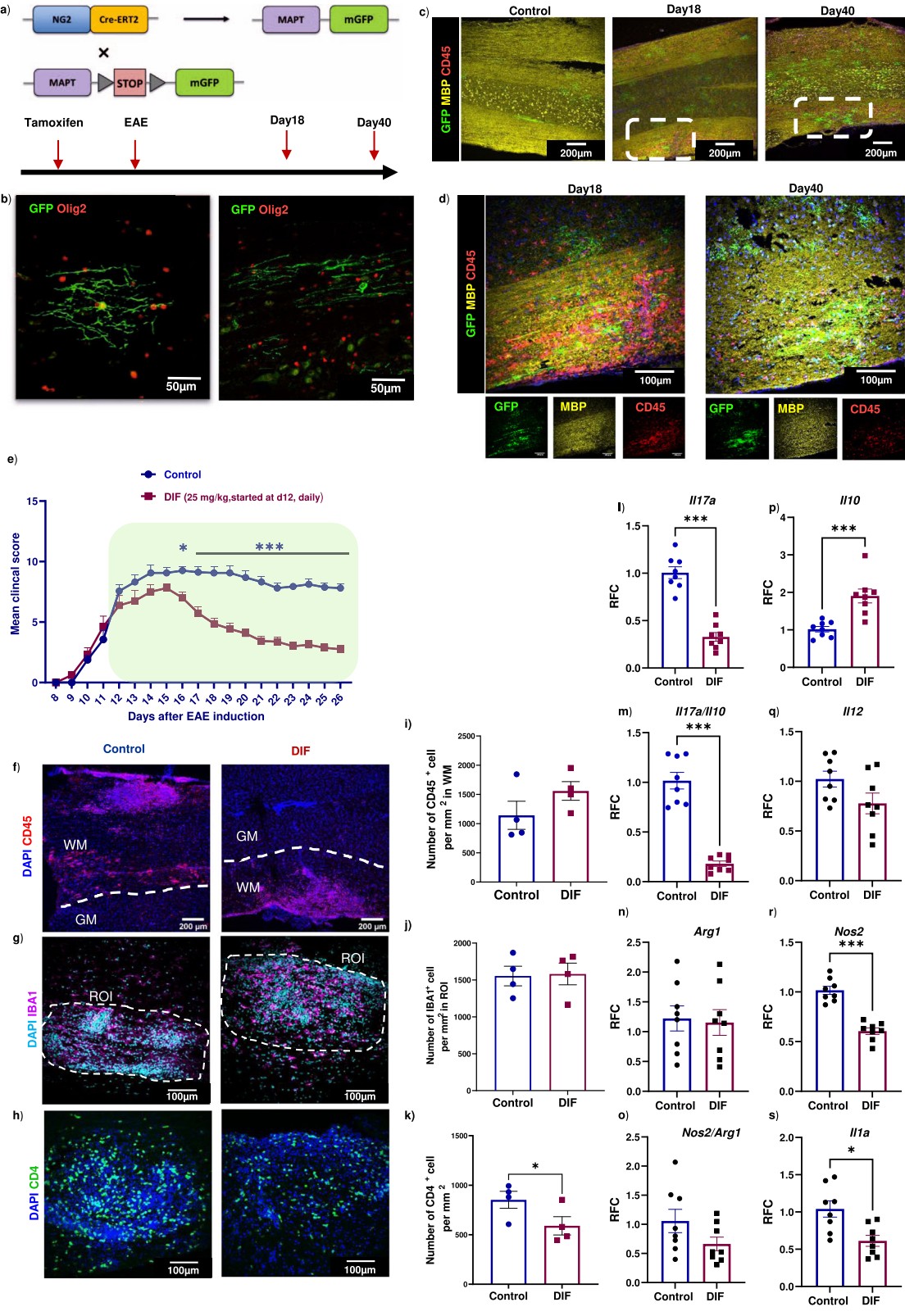

the antibody dilution buffer for 1 h at room temperature. Antibodies were diluted in 0.01 M PBS containing 1% BSA, 0.1%cold fish skin gelatin, and 0.5% Triton-X100. The slides were then washed three more times and then mounted using Fluoromount G (SouthernBiotech). A sample slide stained with only the secondary antibodies and DAPI was used for each experiment as a control for non-specific secondary immunofluorescence.

For MBP staining, sections were also delipidated by sequential wash with 50%, 70%, 90%, 95%, 100%, 95%, 90%, 70%, and 50% ethanol. Then, samples were

rehydrated with PBS for 10 min, and permeabilized with 0.2% Triton-X100 in PBS for 10 min. To remove the GAG chains so as to facilitate the binding of antibody to the core protein of CSPG members (versicans and aggrecan) and also to the stub chondroitin-4-sulfate, chondroitinase ABC (ChABC, Sigma) digestion was done before the blocking step. Slides were incubated with ChABC diluted in PBS (0.2 U/ mL) at 37 °C for 30 min.

The following primary antibodies were used for immunofluorescence microscopy to identify specific targets: versican V0/V1 (Millipore, ab1033),

**Fig. 7 Improved clinical EAE score following difluorosamine treatment in NG2$^{CreER}$:MAPT$^{mGFP}$ mice is associated with the altered balance of immune cell subsets. a** NG2$^{CreER}$:MAPT$^{mGFP}$ mice were used where GFP expression denotes newly formed oligodendrocytes and myelin. **b** Representative images of new myelinating oligodendrocytes stained with GFP (green) and olig2 (red). **c, d** Representative images of longitudinal sections comparing control and peak (Day 18 post-inoculation) or post peak (Day 40) clinical severity of EAE. Stains were GFP (green), MBP for myelin sheaths (yellow) and CD45 (red). Dotted white lines indicate the lesion areas magnified in **d**. Similar results were noted in a separate experiment. **e** NG2$^{CreER}$:MAPT$^{mGFP}$ mice were injected with difluorosamine (DIF, 25 mg/kg) starting three days post clinical onset of EAE, for 15 days (treatment period shown by the green box), and average EAE daily clinical score (mean ± SEM) is shown. $n = 8$ mice in each group from 2 independent experiments of 4 mice. Two-way repeated-measures ANOVA with Sidak's post-hoc test: *$p = 0.03$, ***$p < 0.001$. **f–h** Spinal cords following treatment were immunostained for CD45, IBA1 and CD4. **i–k** Bar graphs (mean ± SEM) of number of CD45$^+$, IBA1$^+$ or CD4$^+$ cells in the white matter or lesion ROI. Data are from one representative experiment of 4 mice in each group, where each dot represents one mouse where a mean of 10 lesions were analyzed per mouse. One-tailed unpaired Student's t test: *$p = 0.04$. This data was reproduced in a second experiment ($n = 4$ mice). **l–s** mRNA expression (mean ± SEM) analyses of lumbar spinal cords comparing the levels of IL-17, IL-10, IL-17/IL-10 ratio, IL-12, arginase-1 (Arg1), inducible nitric oxide synthase (iNOS) and IL-1α between DIF and vehicle treated EAE mice. Data pooled from two independent experiments, $n = 8$ mice, two-tailed unpaired Student's t test; *$p = 0.01$, ***$p < 0.001$. Source data are provided in the source data file.

versican V0/V2 antibody (Millipore, ab1032), aggrecan (Millipore, ab1031), chondroitin-4-sulfate antibody (Millipore, MAB2030) heparan sulfate proteoglycan (Amsbio, 370255), fibronectin (Abcam, ab23750), fibrinogen (Abcam, ab34269), thrombospondin-1 (Abcam, ab85762), myelin basic protein (MBP, Abcam, ab7349), Olig2 (Millipore, ab9610), platelet-derived growth factor receptor α (PDGFRα, R&D Systems, AF1062), adenomatous polyposis coli (APC, Millipore, clone CC-1, OP80), GFP (Aveslab, GFP-1020), ionized calcium-binding adaptor molecule1 (Iba1, Wako, 019-19741), CD3 (Abcam, ab5690), CD4 (BD Pharmingen, 56-0042-82), CD45 (anti-mouse BD Pharmingen, 550539), CD45 (anti-human, Invitrogen, MA5-17687), and versican-V1 (anti-human, Novus biologicals, NBP1-85432).

Images were captured on the Leica TCS SP8 confocal laser scanning microscope and Olympus VS110 Slide scanner. The same laser, gain and offset settings were used for all stained sections within each set of experiments. The z-stacks of confocal images were analyzed with ImageJ (NIH). IMARIS software (Bitplane) was used for 3D rendering of confocal image z-stacks.

For GFP + staining (Fig. 8), the entire white matter of longitudinal thoracic spinal cord section was analyzed. To enumerate the GFP$^+$ areas, two denominators were used. The first involved imaging the whole thoracic spinal cord section and then selecting the white matter region according to MBP staining of an adjacent section; this enabled measurement of the percent area covered by GFP$^+$ profiles in the entire white matter. The second denominator was lesion in the thoracic spinal cord defined by CD45$^+$ accumulation. Each lesion was a region of interest (ROI) and GFP expression was evaluated within each ROI. To obtain the GFP readout, and after setting the color brightness threshold in ImageJ, the Analyze Particles function of ImageJ was used to create a mask and to quantify the percent area occupied by positive signal; the number and average size of particle (positive signal) in each ROI was also captured. Ten to fifteen ROIs were analyzed per mouse. The same color threshold values for setting the positive signal, as well as the size and circularity settings for particle analysis, were used across all samples for each experimental set.

**Immunohistochemistry.** Paraffin sections were deparaffinized and subjected to antibody staining as described elsewhere[45]. Briefly, following deparaffinization, sections were subjected to antigen retrieval in boiling 10 mM sodium citrate buffer (pH 6.0) for 10 min. Once cooled to room temperature, sections were treated to endogenous peroxidase inactivation using 1% H$_2$O$_2$ in methanol. For the detection of versican v1, sections were further enzymatically digested with chondroitinase-ABC (0.2 U/ml, in PBS) at 37 °C for 30 min. Sections were next permeabilized with 0.25% triton-X 100 for 15 min. The sections were incubated with 4% horse serum to block nonspecific binding, then incubated with rabbit anti-human Versican-V1 (Novus biologicals, NBP1-85432) or mouse anti-PLP (Serotec, MCA839G) or mouse anti-BCAS1 (abcam, ab106661) overnight at 4 °C, followed successively by the biotinylated secondary antibody, avidin-biotin complex reagent (Vectastain ABC kit, Vector Laboratories), and diaminobenzidine. The slides were then lightly counterstained with hematoxylin, dehydrated, and mounted. Images were captured on Olympus VS110 Slide scanner.

**Flow cytometry of spinal cord.** Mice were anesthetized, euthanized, and perfused with PBS as described before. The spinal cords were dissected, minced, and enzymatically dissociated by incubating with 2.5 mg trypsin and 5 mg collagenase in 5 ml DMEM media for 20 min at 37 °C. To remove the myelin debris, cells were overlaid on Debris Removal Solution (Miltenyi Biotec) according to the manufacturer's instructions. An average of 2 million cells was isolated from each animal. Cells were seeded in 24 well plates at a density of $1 \times 10^6$ cells in 1 ml of RPMI 1640 medium (Gibco) supplemented with 5% FBS (Gibco) 100 U/ml penicillin, 100 mg/ml streptomycin and activated with cell activation cocktail (BD Bioscience) for 4 h. Cell activation cocktail containing phorbol-12-myristate-13-acetate (PMA), ionomycin, and brefeldin A allows intracellular detection of cytokine. Following T cell

reactivation, cells were assessed for the expression of different T markers using flow cytometry. Cells were washed and then resuspend at $1 \times 10^6$ cells/ml in serum-free staining buffer (BD Bioscience). One μl of Fixable Viability Stain 620 Stock Solution (BD Horizon) was added to 1 ml of cell suspension. The mixture was incubated at 37 °C for 5–7 min and then washed twice with 2 ml of Stain Buffer containing FBS (BD Pharmingen). Fc receptors were blocked using anti mouse CD16/32 (Mouse BD Fc Block, BD Bioscience,1:100) for 30 min away from light at 4 °C. Cells were then immunolabeled with Alexa Fluor 700 anti-mouse CD4 (clone RM4-5, BD Pharmingen), PerCP/Cyanine5.5 anti-mouse CD8a (clone 53-6.7, Biolegend) and APC-eFluor 780 anti-mouse CD3 Monoclonal (clone17A2, eBioscience) for 30 min (4 °C). Next, cells were fixed with 250 μl of fixation buffer (BD Bioscience) at 4 °C in the dark for 20 min. Cells were then washed and permeabilized with 1 ml permeabilization buffer (BD Bioscience). Permeabilized cells were incubated with APC anti-mouse IFN-γ (clone XMG1.2, Biolegend), PE anti-mouse IL-17 (clone TC11-18H10.1, Biolegend), BV421 anti-mouse RORγt (clone Q31-378, BD Horizon) and FITC anti-mouse/human FOXP3 (clone FJK-16s, eBioscience) in the dark for 20 min. Finally, immunolabeled cells were washed and analyzed by a Attune NxT Flow Cytometer (ThermoFisher) equipped with Attune NxT software (v3.1.2) and FlowJo software.

**Antibodies.** Please refer to the Reporting Summary for information on the full list of antibodies used in this study.

**RNA isolation, cDNA synthesis, and real-time PCR.** Total RNA was isolated from lumbar spinal cords using RNeasy Mini kit (Qiagen) according to the manufacturer's instructions and stored at −80 °C. The concentration and purity of RNA were determined by measuring absorbance at 260/280 nm using a nanodrop spectrophotometer (Thermo Fisher Scientific). First strand cDNA synthesis was performed with 1 μg total RNA using miScript II RT Kit (Qiagen) for mRNA expression analyses according to the manufacturer's instructions. Real time PCR was performed using QuantiFast SYBR Green master mix (Qiagen) and Quantitect Primers: Il10 (Qiagen,QT00106169); Il17a (Qiagen, QT00103278); Il1a (Qiagen, QT00113505); Arg1 (Qiagen, QT00134288); NOS2 (Qiagen, QT01547980); Il2 (SA Biosciences, PPM03020E); Actb (Qiagen, QT00095242) and Gapdh (Qiagen, QT01658962) with the following cycling conditions: 95 °C for 5 min, 40 cycles of denaturation at 95 °C for 30 s, annealing at 60 °C for 30 s and extension at 72 °C for 30 s. βactin and Gapdh housekeeping genes were used to normalize mRNA expression. The relative expression levels were analyzed by $2^{-\Delta\Delta ct}$ method.

**Spatial RNA sequencing (spRNAseq).** SpRNAseq was performed using the Visium Spatial Gene Expression platform (10X Genomics). Fresh frozen healthy control tissue (CO36 and CO54) and tissue from two people with MS (MS AB172, MS AB200) were cut into 10 μm sections using a cryostat (ThermoFisher Scientific) then collected on to a Visium Spatial Tissue Optimization Slide and a Visium Spatial Gene Expression Slide with 4 capture areas (1 from each healthy control, 1 chronic active lesion from each MS case). Tissue optimization was performed using the Visium Spatial Tissue Optimization Slide & Reagents Kit, according to the manufacturer's instructions which determined the optimal tissue permeabilization time was 18 min. Brain sections on the Visium Spatial Gene Expression Slide were first stained with hematoxylin and eosin (H&E) according to manufacturer's instructions. Following this, permeabilization, reverse transcription, second strand synthesis and denaturation, cDNA Amplification and QC, as well as library construction and sequencing was performed according to the Visium Spatial Gene Expression User Guide (CG000239). They were loaded at 300 pM and sequenced on a NovaSeq 6000 System (Illumina) using a NovaSeq 200 cycle S1 flow cell. Sequencing was performed using the following read protocol: read 1: 28 cycles; i7 index read: 10 cycles; i5 index read: 10 cycles; and read 2: 90 cycles.

The sequencing depth obtained ranged from approximately 120–142 × 10$^6$ reads per sample. The base call (BCL) files and histology images were processed

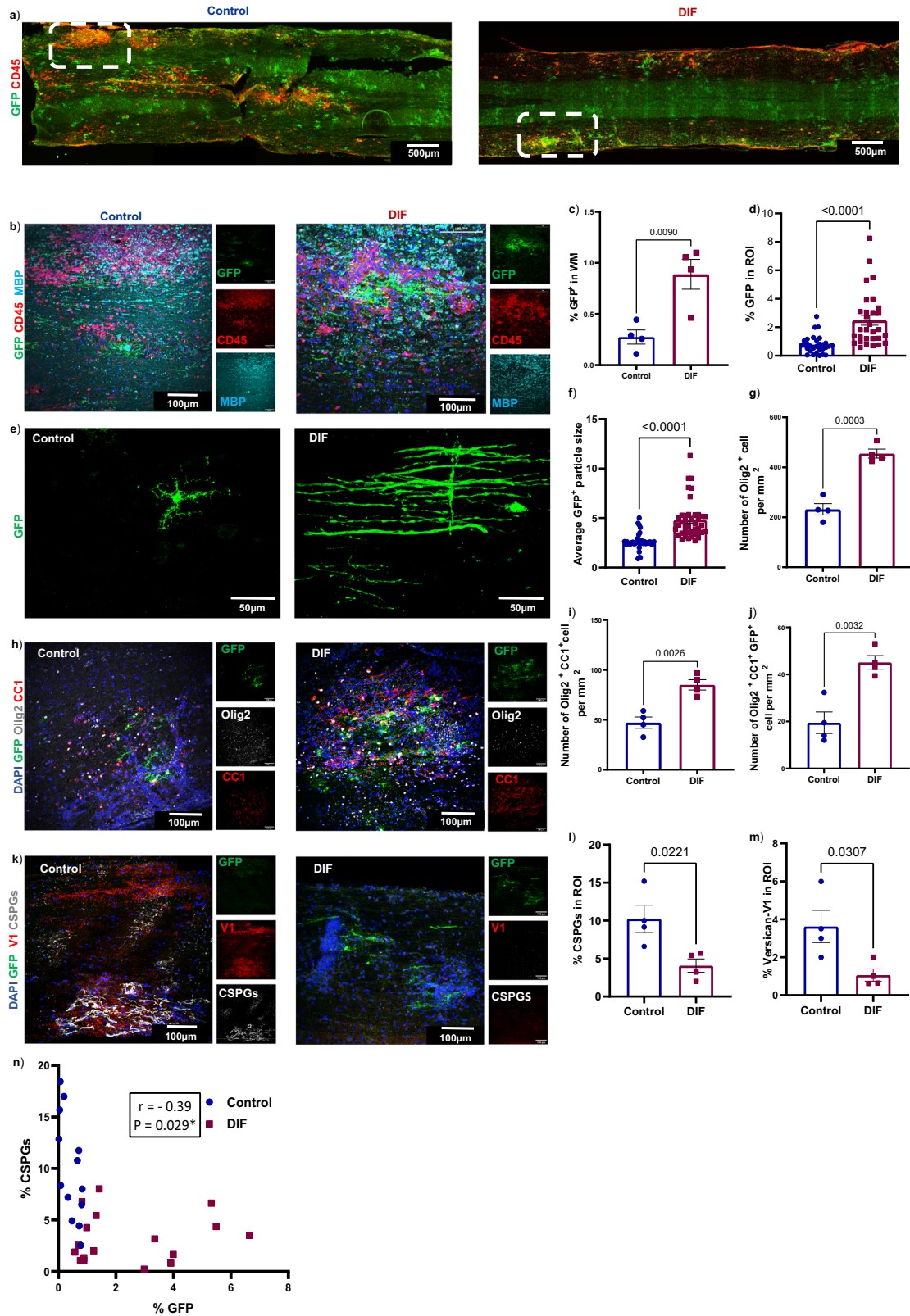

with the Space Ranger software v.1.2, which uses STAR v.2.5.1 for genome alignment, against the GRCh38 human reference dataset. The count files generated for each library were then aggregated with normalization set to 'Mapped'. The aggregated cloupe file was visualized in Loupe Browser. Spatial gene expression data from human MS tissue was visualized using the Loupe Browser 5.01 (10X Genomics). H&E stains were used to define areas of white matter and exclude gray matter in healthy control tissues. In the MS samples H&E was used to define the areas of the inactive core, active rim, and NAWM then the UMAP of the samples

were used to define the exact boundaries. Heatmaps of DEGs between the Control NAWM, MS NAWM, active rim, and inactive core were generated for genes of interest.

## Cell culture

*Mouse oligodendrocyte precursor cells (OPC).* Brains from postnatal day P0-2 mouse pups were isolated and meninges and choroid plexus tissue were removed

**Fig. 8 Difluorosamine reduces versican-V1 expression and promotes presumed remyelination in EAE.** Longitudinal sections of the thoracic spinal cord from EAE NG2[CreER]:MAPT[mGFP] mice following DIF or vehicle treatment were analyzed. **a** Examples of slide scanner images showing lesions defined by CD45[+] accumulation; the dotted rectangles are correspondent with high magnification images in **b** acquired by confocal microscopy (Z-stack). **c** Bar graph ($n = 4$ mice per group) comparing the extent of GFP[+] in the white matter (WM), expressed as % of GFP immunoreactivity of the whole thoracic section. **d** Bar graph of GFP[+] in region of interest (ROI), where each ROI is a lesion defined by area occupied by CD45[+] cell ($n = 30$ ROIs from 4 mice per group). **e** GFP[+] cells. **f** Bar graph comparing average particle size of GFP in each CD45[+] lesional ROI ($n = 41$ ROIs from 4 mice per group. **h** Representative immunofluorescent images (z-stack) labeled for GFP (green), olig2 (white) and CC1 (red). **g, i, j** Bar graphs comparing the number of olig2[+], mature oligodendrocytes (olig2[+]CC1[+]) and GFP[+] oligodendrocytes in the white matter as defined in **c** ($n = 4$ mice). **k** Representative images (z-stack) labeled with GFP (green), versican-V1 (red) and stub chondroitin-4-sulfate (gray). **l, m** Quantification of the percent area occupied by immunoreactive CSPGs or versican-V1 in lesional CD45[+] ROI ($n = 4$ mice). Data presented as mean ± SEM from one experiment, $n = 4$ mice in each group, each dot represents one mouse where a mean of 8 lesions were analyzed per mouse, two-tailed unpaired Student's t test. This data was reproduced in a second experiment. **n** Correlation analysis was performed between percent of GFP[+] and CSPGs[+] in the lesion ROI. Each dot represents one ROI from 4 mice per group. Pearson r −0.39; *$p = 0.029$. Source data are provided in the source data file.

using a dissection microscope. Cortices were removed and dissociated with a digestion cocktail containing papain (1.54 mg/ml, Worthington), DNase (60 µg/ml, Sigma), and L-cysteine (360 µg/ml, Sigma) in a 37 °C water bath for 30 min. Following centrifugation at 300 g for 10 min, cells were resuspended in growth medium and plated in T-75 culture flasks pre-coated with poly-L-lysine (100 µg/mL, Sigma). The mixed glial culture was incubated in DMEM (Gibco) containing 10% FBS, 1% GlutaMAX, 1% sodium pyruvate, and 1% Penicillin-Streptomycin (all from Gibco) at 37 °C, 8.5% CO2 for 9 days. Medium change was performed after 3–4 h, on day 3 and 6. Medium was supplemented with a final concentration of 5 µg/ml insulin (Sigma) from day 6. On day 9, flasks were shaken on an orbital shaker first at 50 rpm for 45 min and then at 220 rpm overnight in a 37 °C, 5% $CO_2$ incubator. Gentle shaking at 50 rpm removed any loosely adherent contaminating cells from the monolayer. Culture medium was replaced with 10–12 ml of fresh mixed glial culture media supplemented with 5 µg/ml of insulin before shaking at 220 rpm that dislodge OPCs and microglia from astrocytes. The supernatant containing OPCs and microglia were collected and added to a 100 mm tissue culture dish. Petri dishes were incubated at 37 °C and 5% $CO_2$ for 30 min, to allow microglia to adhere. Medium containing enriched OPCs was harvested and centrifuged at 300 g for 10 min. Cells were then seeded at a density of $1 \times 10^4$ cells per well in 100 µL of oligodendrocyte differentiating medium (described below) and grown at 37 °C and 8.5% $CO_2$ for 24–72 h. 96-well flat black/clear plates pre-coated with poly-L-lysine was used. Oligodendrocyte differentiating medium was DMEM containing 2% (v/v) B27 supplement (Gibco), 1% (v/v) oligodendrocyte supplement cocktail (see below), 1% (v/v) GlutaMAX™ (Gibco), 100 µM sodium pyruvate (Gibco), 1% (v/v) Penicillin-Streptomycin (Gibco), 50 µg/mL holo-transferrin (Sigma), 5 µg/mL N-acetyl-L-cysteine (Sigma), 50 ng/mL ciliary neurotrophic factor (PeProTech), 10 µg/ml Biotin (Sigma) and 0.01% (v/v) Trace Elements B (Fisher Scientific). Oligodendrocyte supplement cocktail was made from 100 mL DMEM with 1% BSA, 0.6 mg progesterone (Sigma), 161 mg putrescine (Sigma), 0.05 mg sodium selenite (Sigma), 4 mg 3,3′,5-triiodo-L-thyronine (Sigma) and 4mg L-thyroxine (Sigma). The purity of olig2[+] oligodendrocyte lineage cells was over 80% (Supplementary Figure 3A).

For Immunocytochemistry, OPCs were fixed for 10 min at RT using 4% paraformaldehyde, rinsed with PBS and then permeabilized with 0.2% Triton X-100 for 10 min at RT. Odyssey Blocking Buffer (LI-COR) was used for blocking step for 1 h at RT. The primary antibodies including mouse anti-mouse sulfatide O4 (oligodendrocyte lineage cell marker) (R&D) and rabbit anti-mouse MBP (Abcam,) were diluted in Licor blocking buffer. Next, cells were incubated with primary antibodies overnight at 4 °C, then washed 3× with PBS. Secondary antibodies (Jackson ImmunoResearch) and nuclear yellow was added in blocking buffer for 1 h at RT. Cells were finally washed 3× and resuspended in PBS for quantitative fluorescence microscopy analysis by ImageXpress.

*Human oligodendrocytes.* Human brain tissues from patients undergoing surgical resection to treat intractable epilepsy were processed as previously described[14]. Briefly, tissue segments were dissociated using 0.25% trypsin and 20 µg ml[-1] DNase for 1 h at 37 °C and then passaged through a 130 µm pore size nylon filter. Next, the filtrate was centrifuged in 30% percoll for 30 min at 15,000 r.p.m. The viable cell layer was collected and resuspended in Eagle's minimum essential medium (Gibco) supplemented with 5% FCS. Following 48 h of incubation at 37 °C in a 5% $CO_2$ environment, the medium containing floating cells (enriched oligodendrocyte progenitor cells) was collected and plated on poly-L-ornithine-coated 96-well flat-bottom black/clear plates at a density of $5 \times 10^4$ cells per well for 24 h in oligodendrocyte-differentiating medium containing DMEM/F12 containing 1% (v/v) N2 supplement (Gibco), 0.1% (v/v) 3,3′,5-triiodo-L-thyronine (Sigma), 10 nM biotin (Sigma), 100 µg ml[-1] BSA (Sigma), 10 ng ml[-1] PDGF (PeproTech), 1% (v/v) GlutaMAX (Gibco), 100 µM sodium pyruvate (Gibco) and 1% (v/v) penicillin–streptomycin (Gibco). After 24 h of cell attachment, the culture medium was switched to the same medium but lacking PDGF. Cultures were over 75% enriched as monitored by O4 marker. The use of the surgical material for the

current study was approved by The Conjoint Health Research Ethics Board at the University of Calgary.

*CD4[+] T cell isolation and polarization.* Naïve CD4[+] T Cells were isolated from single-cell suspensions of splenocytes using the EasySep Kit (STEMCELL) by negative selection. Unwanted cells were removed with biotinylated antibodies directed against non-naïve CD4[+] T cells (CD8, CD11b, CD11c, CD19, CD24, CD25, CD44, CD45R, CD49b, TCRγ/δ, TER119) and streptavidin-coated magnetic particles. Labeled cells were separated using an EasySep magnet (STEMCELL) without the use of columns. Purified cells were seeded in 24 well plates at a density of $1 \times 10^6$ cells in 1 ml of RPMI 1640 medium (Gibco) supplemented with 10% FBS (Gibco),1%GlutaMAX,1% sodium pyruvate, and 1% Penicillin-Streptomycin (100 mg/ml) (all from Gibco). Plates were coated with ECM proteins (10 µg/ml) for 3 h at 37 °C. Different ECM molecules were used in this study including CSPG (Millipore, CC117), versican-V1[46], thrombospondin (Sigma, ECM002), heparan sulfate (Sigma, H4777), fibronectin (Sigma, F4759) and fibrinogen (Millipore, 341576). The versican-V1 was purified from bovine aorta by a combination of ion exchange and size exclusion chromatography. This preparation contains full length V0/V1 versican with the majority (90%) consisting of V1 as assessed by Coomassie Blue staining and western blots after chondroitinase ABC treated core proteins[46]. Cells were activated with 25 µL pre-washed and resuspended Dynabeads (Gibco) magnetic beads coated with anti-CD3 and anti-CD28 to obtain a bead-to-cell ratio of 1:1. To polarize cells toward different Th subsets different cytokine regimens were used. For regulatory cells, IL-2 (20 ng/ml), TGF-β (10 ng/ml), anti-IFN-γ (10 mg/ml), and anti-IL-4 (10 mg/ml) were added to cells. To polarize cells toward Th17 phenotype, TGF-β (2 ng/ml), IL-6 (50 ng/ml), IL-1β (20 ng/ml), IL-23 (50 ng/ml), anti-IFN-γ (10 mg/ml) and anti-IL4 (10 mg/ml) were used. For polarization toward Th1 phenotype, IL-2 (2 ng/ml), IL-12 (50 ng/ml), and anti-IL-4 (10 mg/ml) were added. All cytokines were purchased from Peprotech and antibodies were from Biolegend.

To block some receptors potentially interacting with CSPGs, ILP (NH2-GRKKRRQRRRCDLADNIERLKANDGLKFSQEYESI-NH2) and ISP (NH2-GRKKRRQRRRCDMAEHMERLKANDSLKLSQEYESINH2) peptides against LAR and PTPσ, respectively, were added to cells[18]. Intracellular Mu peptide (IMP, NH2-LLQHITQMKCAEGYGFKEEYESGRKKRRQRRRCNH2, CS Bio Co) was used as a control peptide. Cells were pretreated with 2.5 µM of IMP, ILP and ISP for 30 min and then transferred to CSPG-coated wells. In a parallel experiment, cells were pretreated with 50 µg/ml blocking antibodies against mouse integrin-beta3 (CD61, Invitrogen), mouse integrin-beta1 (CD29, Invitrogen) or mouse integrin-beta6 (Millipore). Cells were incubated in a humidified CO2 incubator at 37 °C. After 96 h, the beads were removed and conditioned medium was collected for analysis with IL-17 ELISA (Invitrogen), which was performed according to manufacturer's instructions.

To perform in vitro MOG stimulation experiments, isolated T cells from 2D2 mice were activated with MOG-loaded mature dendritic cells (ratio: 2:1) and then polarized to Th17 cells as described before.

Following T cell reactivation, cells were assessed for the expression of Th1, Treg, and Th17 cells differentiation markers using flow cytometry as described above. Cells were immunolabeled with Alexa Fluor 700 anti-mouse CD4 (clone RM4-5, BD Pharmingen), PerCP anti-mouse/human CD44 (clone IM7, Biolegend), Brilliant Violet 510 anti-mouse CD62L (clone MEL-14, Biolegend), APC-eFluor 780 anti-mouse CD25 (clone/l PC61.5, eBioscience), APC anti-mouse IFN-γ (clone XMG1.2, Biolegend), PE anti-mouse IL-17 (clone TC11-18H10.1, Biolegend), BV421 anti-mouse RORγt (clone Q31-378, BD Horizon), FITC anti-mouse/human FOXP3 (clone FJK-16s, eBioscience). Immunolabeled cells were analyzed by a Attune NxT Flow Cytometer (ThermoFisher).

*Dendritic cells.* Bone marrow-derived dendritic cells (BMDCs) were prepared from femurs and tibiae of C57/BL6 mice[47]. Extracted bone marrow cells were cultured at a density of $1.5–3 \times 10^6$ cells per ml in RPMI 1640 culture medium containing 10% FBS, 100 U/ml penicillin, 100 mg/ml streptomycin, and 200 ng/ml recombinant

FLT3L (Biolegend) for 8–10 days. Cells were seeded in control or CSPG-coated (10 μg/ml) 24-well plates at a density of $1 \times 10^6$ cells and treated with 50 μg/ml MOG (synthesized by Protein and Nucleic acid facility, Stanford University) in the absence or presence of lipopolysaccharide (LPS, Sigma,10 ng/ml) for 12 h. Different types of DCs and maturation markers were analyzed by flow cytometry using Super Bright 702 anti-mouse CD11b (clone M1/70, eBioscience), APC anti-mouse CD80 (clone 16-10A1, Biolegend), PE anti-mouse CD86 (clone GL-1, Biolegend), PE/Cyanine7 anti-mouse CD11c (clone N418, Biolegend), Alexa Fluor 700 anti-mouse F4/80 (clone BM8, Biolegend), FITC anti-mouse I-A/I-E (clone M5/114.15.2, Biolegend), PerCP/Cyanine5.5 anti-mouse CD317 (BST2, PDCA-1) (clone 927, Biolegend), APC 750 anti-mouse CD24 (clone M1/69, Biolegend), V450 anti-mouse IL-12 (p40/p70) (clone C15.6, BD Horizon).

*Live imaging of OPC - Th17 cell co-culture.* Mouse OPCs grown in 96-well flat bottom black/clear plates ($1 \times 10^4$ cells per well) were incubated with 1 μM Calcein-AM (ThermoFisher Scientific). Th17 cells were reactivated on day 3 using anti-CD3 (BD, 0.5 μg/ml) for 2 h and then labeled with 10 μM Deep Red cell tracer (Invitrogen) for 15 min at 37 °C. Labeled Th17 cells ($2 \times 10^4$ cells per well) plus Hoechst 33342 NucBlu Live Ready Probes Reagent (Thermofisher Scientific) and propidium iodide (Thermofisher Scientific) were added to OPCs. Cells in the same FOVs were imaged at 2 h intervals using the ImageXpress Micro XLS High-Content Analysis System, for a total of 12 h.

*ImageXpress acquisition and MetaXpress analysis.* Labeled cells in 96-well flat bottom black/clear plates were imaged with ImageXpress® Micro Cellular Imaging and Analysis System (Molecular Devices). For each well, twelve images (field of views;FOVs) were acquired for quantitative analysis. Images were then processed with MetaXpress High-Content Image Acquisition and analysis software (Molecular Devices). "Multiwavelength cell scoring" software module was used to quantify the cell survival and also cell number for a particular marker. OPC outgrowth was measured by the MetaExpress® "neurite outgrowth" software module which quantify cell branches. Data from the 12 images were averaged to a single data point per well, with four well replicates per treatment.

*Statistics.* Microsoft Excel (Version 2201 Build 16.0.14827.20198) was used for collating data, All graphs were generated using GraphPad Prism 9.0.2 (LaJolla). Where multiple groups were compared, one-way ANOVA with Bonferroni multiple comparison test was used. For comparisons between two groups, significance was determined by unpaired two-tailed Student's *t*-tests. EAE disease scores were analyzed with two-way repeated-measures ANOVA with Sidak's post-hoc test. The correlation analysis (Pearson's correlation coefficient) was performed between the levels of 2 different markers (GFP and CSPGs). Kolmogorov-Smirnov test was applied to verify normal distribution of data. *p* values below 0.05 was considered statistically significant shown by asterisks in the figures ($*p < 0.05$, $**p < 0.01$, $***p < 0.001$). All values are shown as mean ± SEM.

**Reporting summary**. Further information on research design is available in the Nature Research Reporting Summary linked to this article.

## Data availability

Source data are provided with this paper. All datasets generated and/or analyzed during the current study are available from the corresponding author on reasonable request. There are no restrictions on data availability. Raw spatial RNA sequencing data are available at the NCBI Sequence Read Archive with the BioProject accession number: PRJNA734097.

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

## Acknowledgements

We thank the Hotchkiss Brain Institute Advanced Microscopy Platform facility for microscopy and image analysis platforms. We thank the UK Multiple Sclerosis and Parkinson's Tissue Bank at Imperial College, London, and Dr. Djordje Gveric for the tissues in Fig. 1a, b. We thank Dr. Alex Prat (University of Montreal) for the samples in Supplementary Fig. 1. This work was funded by operating grants from the Multiple Sclerosis Society of Canada (MSSC) and Canadian Institutes of Health Research (CIHR) to VWY (grant number 3527 and FDN 167270, respectively). SG acknowledges postdoctoral fellowship support from the Harley N. Hotchkiss Postdoctoral Fellowship, MSSC and CIHR. R.J. and D.K.K. acknowledge postdoctoral fellowship support from MSSC. B.L. gratefully acknowledges studentships from the Alberta Graduate Excellence Scholarship and MSSC. V.W.Y. acknowledges salary support from the Canada Research Chair (Tier 1) program.

## Author contributions

S.G. designed the project, performed the majority of experiments, analyzed the results, and wrote the first draft of the paper. E.J. performed some of the oligodendrocyte culture studies. R.J. contributed SpRNAseq data. B.B. and C.L. helped with data analyses. B.L. performed lysolecithin surgeries. S.S. stained M.S. samples for BCAS and versican-V1. D.K.K. provided data of MS brain tissues. Y.D. helped with 2D2 T cell cultures. G.J.S., E.M.S., and J.G. characterized and provided BCAS-positive MS brain tissues. S.K. contributed to the blocking peptide experiments. T.W. purified and provided versican-V1. P.Z. and C.C.L. designed and synthesized difluorosamine. V.W.Y. supervised the study, provided operational support, and edited and finalized the manuscript. All authors reviewed and edited the manuscript.

## Competing interests

PZ, CCL and VWY have a patent application (USA Patent application number 17/059,318) pending on the use of difluorosamine in multiple sclerosis. Other authors declare no competing interests.
