## [Peer Review File · Nature Communications]

Reviewers' Comments:

Reviewer #1:

Remarks to the Author:

The failure of remyelination in the context of multiple sclerosis and human neurological disease contributes to disease progression and disability. Following demyelination, the pathological extracellular matrix (ECM) exerts powerful influences on both neuroinflammatory processes and myelin regeneration. Among the various components of the ECM, chondroitin sulfate proteoglycans have emerged as key inhibitory components that limit myelin regeneration. CSPG core proteins represent a large class of cell-surface and shed glycoproteins that are expressed by many cell types within the CNS.

Previous work by the group identified an increase in versican V1 mRNA at the peak of disease in EAE and that total versican protein and cleavage were also increased. Versican protein was associated with perivascular cuffs in EAE and MS brain tissue. They have also showed that fluorosamine, an inhibitor of chondroitin sulfate biosynthesis, reduced infiltration of CD45+ cells and reduced EAE severity.

In this manuscript, Ghorbani et al. the authors extend upon these data to examine the role of CSPG in polarization of Th17 cells and the show that CSPG-polarized Th17 exert more cytotoxicity on OPCs and oligodendrocytes. The authors also demonstrate that DIF, a small molecule epimerase inhibitor that prevents the biosynthesis of chondroitin sulfate and heparan sulfate, induces remyelination in the context of EAE and that this is associated with reduced Th17 infiltration and modulation of the associated neuroinflammatory processes. Overall, this paper provides novel insight into the roles of CSPG and versican V1 in the context of autoimmune-mediated demyelination.

Strengths of this manuscript include the high quality of the data, rigorous experimental design, and statistical analysis. The evidence of an inverse spatial correlation of versican V1 and either lower densities of oligodendrocytes in MS lesions or oligodendrocyte generation in EAE is a compelling association but the experiments described lacks a formal proof that versican V1 is the functional CSPG in vivo. In vitro CSPG exposure causes Th17 cells to exert a greater cytotoxic effect on mixed glial culture resulting in the numbers of surviving oligodendrocytes being significantly reduced. The effects of DIF in EAE are potent demonstrating that DIF treatment significantly alters the number and polarization of CD4+ cells in spinal cord and peripheral lymph nodes consistent with a diminution of pro-inflammatory IL-17 cells. The use of the NG2creER:MAPTmGFP mouse provides a reliable measure of oligodendrocyte generation and a useful surrogate for remyelination and demonstrates that DIF treatment improves oligodendrocyte generation in EAE.

There are some specific points that should be addressed in a revision:

1. The authors describe the inverse association of versican V1 levels with the number of BCAS1+ oligodendrocyte lineage cells in MS lesions. BCAS1+ is expressed at high levels by immature oligodendrocyte (iOLs) but is also expressed by OPCs, representing ~20% of the total BCAS1-expressing pool. This should be clarified as unless cell morphology was also considered, BCAS1+ cells in the MS tissue may well represent OPCs as well as iOLs.
2. There appears to be no methodological description of the source and preparation of human OPCs described in Figure 1. The authors provide quantitative data on human OPCs and their response to culture on versican substrate. In addition to the number of O4+ cells, it would be useful to know the proportion of cells which express O4. To help characterize the human OPC culture in general, the proportion of Olig2-expressing oligodendrocyte lineage cells in these cultures should also be reported.
3. The authors note difficulty with low sensitivity of the spRNAseq. Perhaps, the use of RNAscope might be useful alternative technique to look at specific Tcell markers.
4. Figure 2: While of very high quality, the demonstration of versican V1 expression in CD45+ cells do not seem to add much more conceptually than previously published in Brain. Having said that,

the analysis of other ECM molecules such as versican V2 and aggrecan provide additional context and point toward the relatively specificity of versican V1 elevation in EAE.

5. In Fig. 4, the EAE clinical data suggests a cohort specific difference between control and DIF groups prior to DIF treatment. It appears that animal group assignment was not appropriately randomized on the day of induction. While the effect of DIF in Fig 4 is relatively modest, the data presented in Fig 6 is far more convincing and somewhat alleviates these concerns.

6. It is unclear how peak EAE was determined in each cohort and why DIF treatment was initiated at 12 days in Fig 4 and 13 days in Fig 6.

7. Figure 5 shows that CSPG treatment of Th17 cells induces a more cytotoxic effect on OPCs. Can this effect be replicated by just incubation with versican V1? This would further support the premise that the relevant core protein is versican vs. another CSPG. Likewise, would versican V1-depletion prevent these effects in vitro? Additional experiments in this model could further define the relative importance of versican V1.

8. DIF and fluorosamine are fluorinated analogous of GlcNac and act by reducing the production of chondroitin sulfate side chains by inhibiting 4-epimerase. This compound will modulate both heparan sulfate and chondroitin sulfate GAG biosynthesis. Given that HSPGs have been recently implicated in contributing to the inhibitory ECM environment following demyelination (Saraswat et al., Nat Comm 2021), the interpretation that DIF acts solely via CSPG inhibition may not be justified in the absence of other data. Does DIF treatment alter HSPG sulfation, i.e. 10E4 staining? (i.e. Fig. 2I/P in the context of DIF treatment)

9. Although assumed to be nuclei (DAPI?) there is no mention of NY (Fig 6. Panel G) in legend or methods.

10. Figure 7. The use of the stub chondroitin-4-sulfate antibody is not described in the methods. Are all tissues treated with chondroitinase ABC prior to staining? If so, the reduction in versican V1 core protein staining following DIF treatment suggests that appropriate GAG side changes alter the stability and/or proteolytic degradation of versican V1. Perhaps this could be discussed in the manuscript.

Minor:

11. Introduction. Enhanced remyelination was also noted following chondroitinase treatment in Segel et al., Nature 2019.

Reviewer #2:

Remarks to the Author:

Despite a considerable literature on changes in the extracellular matrix in MS, we still have a very poor understanding of how this matrix might enhance or perturb the parallel processes of damage and repair occurring in the disease. This paper examines one particular group of extracellular matrix components, the chondroitin sulphate proteoglycans and in particular versican. The paper starts with an interesting analysis of versican expression and oligodendrocyte numbers in MS lesions, showing an inverse correlation, and then moves onto experiments examining the effect of versican on TH 17 T cell polarisation/secretion and oligodendrocyte maturation. Then, using DIF to inhibit proteoglycan synthesis in a loss of functional approach the authors show a reduction in the extent of new oligodendrocyte formation in the EAE model of acute inflammatory demyelination.

The results are interpreted show an effect of versican both T cell polarisation and oligodendrocyte formation in vitro and in vivo. As such the paper represents a potentially valuable addition to the literature. However even if one accepts that the changes in T cell numbers are biologically significant (and see below) the key question is the relative contribution of the T cell changes and oligodendrocyte differentiation effects in the beneficial effects of DIF. This could surely be examined using the methodologies the authors have available; a comparison of the EAE and LPC models of demyelination following DIF administration. In the latter T cell polarisation is not

involved in the generation of damage, and so comparisons would allow the contribution of the two possible effector mechanisms to be dissected apart

In addition;

In the experiments examining oligodendrocyte differentiation, versican reduces cell adhesion as the authors show. This would have significant secondary effects on morphological differentiation, making the latter experiments hard to interpret. Could the authors provide a uniformly adhesive substrate and assess the effect of added matrix in the culture medium?

The conclusion from the MS lesions that high areas of versican expression correlate with low numbers of new oligos requires a quantitative analysis (of ECM expression) rather than the qualitative approach taken in the manuscript

Figure 2D is confusing in that the dashed box on the left-hand panel doesn't correspond to the right-hand panel. Why does the versican labelling in the left-hand panel appear to be intracellular?

In figure 3 the changes in the T cell populations are very modest, raising the question as to whether this would have a significant biological effect. Equally in figure 4 the effect size in vivo is very small-about 4%. How do the authors conclude such small effects will have a major effect on myelin repair? The toxic effects of treated T cells in figure 5 is very interesting, but would the very small changes in population numbers seen in the earlier studies be sufficient, on the basis of data in figure 5, have an effect on oligodendrocyte numbers in vivo?

In Figure 6 the affect of DIF on EAE severity is profound whilst in figure 3 it is marginal. Why are they so different?

Figure 7, which illustrates the key experiment of the paper, requires a quantitative analysis of cell numbers and re-myelination-the latter ideally by electron microscopy or by immunofluorescence to count the formation of new sheaths

The conclusion at the end of the results that "particularly versican" reduce neuro inflammation and promoted remyelination is not justified, given that the key experiments used a broad inhibitor of proteoglycan synthesis

Reviewer #3:

Remarks to the Author:

The current manuscript, "Versican impairs remyelination by inhibiting oligodendrocytes and promoting T helper 17 cytotoxic neuroinflammation", aims to link several key observations about the inhibitory nature of CSPGs on remyelination. This area of work has potential impact for targeting and developing therapeutics for Multiple Sclerosis. This manuscript adds several points that were open questions from their previous studies of Keough et al. Nat Comm publication in 2016 and Stephenson et al. 2018 in Brain. Here the authors further investigate the mechanisms underlying fluorosamine-mediated CSPG inhibition to promote myelin restoration. They extend their EAE studies by taking advantage of EAE combined with a recent strategy to label newly formed oligodendrocytes. This enables the authors to distinguish newly formed oligodendrocytes from existing oligodendrocytes. Importantly, this allows the authors the ability to assess immune cell-mediated demyelination as well as remyelination. Here the authors show the presence of a specific CSPG, VersicanV1, within lesions in patient samples, matching that to what is seen in EAE lesions. Further, the authors provide links to the T cell immune in response to Versican-V1 and CSPGs in vitro and CSPGs during remyelination.

The authors showcase a wide variety of complimentary in vitro and in vivo experimentation to examine the effects of CSPGs on T cell populations and EAE. I came away with the impression that the authors put in a lot of effort into this work. After reading the manuscript, I also had the impression that this is an extension of the authors previous work, Keough et al. 2016 Nat Comm and Stephenson et al. 2018 Brain, examining CSPGs and inhibiting CSPGs to promote

remyelination. The new findings are interesting next steps to these previous publications. The key new findings 1) relate Versican-V1 levels and pre-myelinating cell densities in human post-mortem tissue; 2) demonstrate that Versican-V1 and CSPGs impact T cells; 3) demonstrate that CSPG synthesis inhibition reduces IL17/Th17 during EAE. Their previous work had demonstrated Versican-V1 levels correlate with EAE lesions and that CSPG inhibition reduced severity of EAE (which correlated with reduced Versican mRNA levels from EAE tissue samples), as well as enhancing new mature OLs in LPC remyelination models.

There are several major and minor points regarding the manuscript that should be addressed, however, by the authors. I outline these by topic, as follows:

Authors' Conclusion1: versican-V1 is associated in lesions of MS with reduced remyelination One of the strengths to this work is the examination of post-mortem tissue samples from control and MS patients (Fig1A-E). As one of the key results, I would like to see the following clarifications to enhance the conclusions:

1. Have the example lesions been categorized? Could their categorization be labelled next to the images, please? It is unclear from the figure why certain regions were selected. Are you making conclusions about only active lesions?

2. On a related note, I am still unclear about whether Versican-V1 is primarily associated with demyelinating lesions, active vs chronic active lesions, or inactive lesions. Were the lesions only selected based upon LFB staining or were other classifications used (e.g., criteria that indicate active, inactive demyelinated, chronic active)? Do the Versican-V1 low vs high correlate with lesion type – inactive demyelinated, active demyelinating, chronic active, remyelinating, for example? If the authors have that information, that would also help paint the picture of how the Versican-V1 low vs high links (if at all) to lesion type. In Figure 1A, B the authors indicate looking at active lesions correlates with high V1, whereas the spRNASeq data indicates elevated levels of versican with inactive demyelinated lesions. Commenting further on this would help readers understand more clearly whether V1 levels may be relevant to demyelination and/or remyelination potential.

3. It looks like MS-352 and MS-230 have high versicanV1 associated with CD45+ intensities, but MS-163 has high versicanV1 elsewhere in the brain section. Could the authors clarify these different observations?

4. Could the authors denote different patient samples within their graph Fig1E (either by color-coding the dots or by different shapes)? That would help with the interpretation of how consistent these correlations were between different individuals (as opposed to different lesions).

Authors' Conclusion2: V1 affects adhesion of OPCs and branched morphology of OLs

The authors use isolated mouse OPC cultures and human OPCs to examine direct effects of Versican-V1 on the OL lineage cells. While this is an important experiment, the results leave the impression that Versican-V1 does not have strong effects directly on the OPC/OL cells. There are a few points I'd like the authors to address:

1. Can the authors comment on the discrepancy in protein concentration between mixed CSPGs and purified Versican-V1? I would predict that if Versican-V1 was a main player in directly affecting OL biology, that Versican-V1 would be more potent than the mixture of CSPGs. The results show the opposite, indicating to me that CSPGs collectively are much better at blocking OL branching rather than Versican-V1 itself.

2. What is the control being used? Previous work (Keough et al. 2015) showed that the decreased adhesion of OLs to CSPGs was due to the charge of the surface, not specific to CSPGs per se (Poly-arg peptide+CSPG coating neutralized the effects on OPC adhesion). For your Versican-V1 experiment assessing adhesion (Fig1I), did you similarly control for non-specific effects of surface charge?

3. If pre-myelinating cells are reduced with Versican-V1 in MS patient lesions, one would predict that V1 inhibits OPC differentiation, prevents recruitment of new OLs, or does not support OL

survival. Could the authors clarify how the in vitro experiment assessing branching of OL cells links to these predictions. An assessment of differentiation and/or cell survival would be more intuitive (not required but please comment/clarify).

4. While Versican-V1 at higher doses is quantified to affect outgrowth, the image in Fig1F is difficult to interpret. Could a higher magnification be shown of the cellular morphology?

5. The image in Fig1I with human cells looks like the cells are branched with Versican treatment. Is branching similarly affected for both mouse and human cells?

Authors' Conclusion3: Versican-V1 increases Th17 polarization

In contrast to effects on OPCs/OLs, the impact of Versican-V1 treatment and CSPG inhibition seems clearer for T cells (Figs3, 4). The specific role of versican-V1 vs other CSPGs was compared, which was great. Versican-V1 rather than other ECM found in the lesions (Fig2) enhanced Th17 production.

1. Could the author's note the concentration of Versican-V1 and other recombinant proteins being used in these experiments to allow easier evaluation of the findings, please?

2. Could the authors clarify what is meant by "Data are presented as mean \pm SEM (n=3-4) and are from a single experiment representative of three independent experiments"? What is the "unit" for the "n" here (e.g., separate cell isolations from different mice)? Are data points shown from the three independent experiments or from one single "representative" experiment? Please expand the detail on this throughout the figure legends (also same statement in other legends).

3. While not necessary, it would have been nice to see the experiments in Fig3D-G done with purified Versican-V1, rather than mixed CSPGs. That would have made a more cohesive narrative for the novel conclusions regarding Versican-V1.

Authors' Conclusion4: cell polarization affect toxicity to OPCs

1. Figure 5B: To my eye the number of lymphocytes between the column of Th17 and CSPG-exposed Th17 cells looks greater in the CSPG-exposed Th17 cell condition. Where the numbers of Th17 cells normalized between the conditions?

2. Although not necessary, again examining effects directly by Versican-V1 rather than CSPGs would have been even better.

3. Was this experiment only conducted once with 4-6 wells per condition? I am confused by the legend "one biological replicate representative of three independent experiments". How is it representative of three independent experiments? Was each well a separate cell isolation from different animals?

4. What is the relationship between Fig 5B and 5D (i.e., how was cell viability/toxicity actually measured)? Did the authors quantify the fraction of O4+ or O4+MBP+ cells that label with propidium iodide and/or loss of calcein AM? Was cell number per field of view the only measurable outcome in Fig5D? Please clarify.

Author's Conclusion5: Difluorosamine improves EAE clinical score, altering immune cells and improving remyelination

The demonstration of altering Th17 and IL17 with difluorosamine was an important component to this manuscript. While prior work from their lab showed improved EAE clinical score, this goes the next step to identify changes that may underlie this and raise a new hypothesis about CSPG activation of Th17 cells during de/remyelination.

1. In Fig6: are the numbers of immune cells affected? How were the numbers assessed; some cell type markers are shown as % in WM or ROI and others as number per mm². It is therefore difficult to know exactly how this was assessed. Were statistics conducted to compare between control and DIF treatment. At a glance, it looks like %CD45+ in WM is increased in DIF treatment.

Is it possible to assess CD4+ cell distribution in the EAE lesions, rather than solely by FACs in Figure 4. That would give a better link between T cells and Th17 cells in relation to demyelination/remyelination.

2. In Fig 7: What time point is being assessed? It does not state it in the figure legend and/or figure itself. In DIF treatment, the appearance of CD45+ areas is reduced (Fig 7A) in this example. This was used to define lesions. Therefore, I am unclear selected areas
How many mice were assessed for this experiment?

3. The images in Figure 7I are striking, with a clear difference in sheath-like extensions in the DIF treated example. The quantification was less clear; for the quantification in part J, what is meant by GFP+ particle size? How was the imaging and analysis done for Figure 7? Please detail further how areas were selected and how 'particle' size was measured in the methods. Also, what are the n values used for the statistics; e.g., in J, is each dot representing a single imaging 'field' being treated as a 'n' for the statistics? This image is the best evidence given for remyelination, but quantification of remyelination has not been demonstrated. Instead, the current evidence demonstrates an increase in newly formed oligodendrocytes with processes. I would be hesitant to claim "higher content of new myelin" (line 360) without demonstration of node formation and/or electron microscopy of myelin sheaths. While I don't expect additional experiments per se, the text should be modified so that conclusions about myelin are not made.

Overall Authors' Conclusion: "Our results identify versican-V1 as a therapeutic target in inflammatory demyelinating lesions to reduce Th17 polarization and its toxicity to OPCs, and to promote remyelination in the milieu of ongoing adaptive and innate neuroinflammation."

My recommendation would be to modify the title to reflect the direct findings from this work. While all the evidence points to an inverse correlation of versican-V1 levels with pre-myelinating oligodendrocytes in lesions, and the ability for versican-V1 to enhance Th17 cell polarization, there are a few outstanding experiments to directly test the connection between these elements. The authors utilized many approaches that yielded results leading to the new hypothesis that versican V1 promotes Th17 cytotoxic activity, which in turn decreases OL survival during remyelination. I would argue that this is currently a hypothesis supported by the data rather than a concrete conclusion. Independently, the experiments have demonstrated that 1) versican V1 promotes differentiation of Th17 cells, 2) CSPGs promote Th17 IL-17 levels, 3) one effect of difluorosamine treatment in EAE is a decrease in Th17 IL-17 levels, 4) CSPG-exposed Th17 cells decrease OPC and OL cells in co-culture. There remain outstanding questions to directly test and bridge these conclusions, namely: whether Th17 cells are found in Versican-V1 lesions, whether blocking Th17/IL17 during remyelination would enhance myelin repair, whether decreasing Versican-V1 specifically (rather than CSPGs generally) would decrease Th17/IL17 during remyelination and correspond to an increase in OL number as well as increased myelin repair (rather than solely new OL formation). While I do not expect the authors to conduct these experiments at this time, these would be considered important to really test the hypothesis. Therefore, it would be appropriate to modify the title and text to reflect this as a hypothesis drawn from the conclusions of the work rather than final conclusion itself.

Other points:

1. MAJOR POINT: The statements in the figure legends about n values, biological replicates, vs independent experiments need greater clarification. The addition of units would be extremely helpful. For example, instead of "n=3", please state if "n=3 mice" or "n=3 lesions". Also, it was unclear what was meant by phrasing such as "single experiments representative of three independent experiments". This prevents appropriate evaluation of the results. Please give greater detail for each experiment shown.

Other examples in the figure legends:

a. Fig S2: there is currently no mention of the biological replicates/independent experiments conducted

b. Fig S3: n = 5 wells. Each well is a technical replicate rather than a biological/independent experiment. If so (only technical replicate), statistical testing is not appropriate to draw a conclusion about the biology.

2. The authors have forgotten to include the methods regarding culture of human OPC/OLs.

3. This is stylistic, but I somewhat struggled with the flow/progression of the narrative starting from Versican-V1 in MS tissue followed later by experiments that more broadly examined CSPGs in EAE immune response and remyelination. It made me wonder during all the later figures why Versican-V1 wasn't the focus of experiments (instead of CSPGs broadly). My suggestion would be to write the manuscript in the order starting from the EAE model of remyelination with CSPGs – the effect on immune cells, then myelin, then focus in on T cells with CSPGs, how that narrowed down to Versican-V1 affecting T cell Th17 populations, and Versican-V1 on OPCs, then MS lesions with Versican-V1.

4. In the "Live imaging of OPC-Th17 cell co-culture", 10,000 mOPCs are plated in a 96well, which is a reasonable cell density. However, under "Mouse oligodendrocyte precursor cells (OPC)" line 842, it states cells were plated at 100,000 cells per 96 well. That would be extremely dense. Is that an error?

5. What does RFC stand for in Figure6? It may not be evident to your readers.

Rebuttal to Reviewers

NCOMMS-21-22835, " Versican impairs remyelination by inhibiting oligodendrocytes and promoting T helper 17 cytotoxic neuroinflammation "

In response to Reviewer 1:

“The failure of remyelination in the context of multiple sclerosis and human neurological disease contributes to disease progression and disability. Following demyelination, the pathological extracellular matrix (ECM) exerts powerful influences on both neuroinflammatory processes and myelin regeneration. Among the various components of the ECM, chondroitin sulfate proteoglycans have emerged as key inhibitory components that limit myelin regeneration. CSPG core proteins represent a large class of cell-surface and shed glycoproteins that are expressed by many cell types within the CNS.

Previous work by the group identified an increase in versican VI mRNA at the peak of disease in EAE and that total versican protein and cleavage were also increased. Versican protein was associated with perivascular cuffs in EAE and MS brain tissue. They have also showed that fluorosamine, an inhibitor of chondroitin sulfate biosynthesis, reduced infiltration of CD45+ cells and reduced EAE severity.

In this manuscript, Ghorbani et al. the authors extend upon these data to examine the role of CSPG in polarization of Th17 cells and the show that CSPG-polarized Th17 exert more cytotoxicity on OPCs and oligodendrocytes. The authors also demonstrate that DIF, a small molecule epimerase inhibitor that prevents the biosynthesis of chondroitin sulfate and heparan sulfate, induces remyelination in the context of EAE and that this is associated with reduced Th17 infiltration and modulation of the associated neuroinflammatory processes. Overall, this paper provides novel insight into the roles of CSPG and versican VI in the context of autoimmune-mediated demyelination.

Strengths of this manuscript include the high quality of the data, rigorous experimental design, and statistical analysis. The evidence of an inverse spatial correlation of versican VI and either lower densities of oligodendrocytes in MS lesions or oligodendrocyte generation in EAE is a compelling association but the experiments described lacks a formal proof that versican VI is the functional CSPG in vivo. In vitro CSPG exposure causes Th17 cells to exert a greater cytotoxic effect on mixed glial culture resulting in the numbers of surviving oligodendrocytes being significantly reduced. The effects of DIF in EAE are potent demonstrating that DIF treatment significantly alters the number and polarization of CD4+ cells in spinal cord and peripheral lymph nodes consistent with a diminution of pro-inflammatory IL-17 cells. The use of the NG2creER:MAPTmGFP mouse provides a reliable measure of oligodendrocyte generation and a useful surrogate for remyelination and demonstrates that DIF treatment improves oligodendrocyte generation in EAE.”

We thank the Reviewer for the complimentary comments and for the very helpful critiques of this manuscript.

*“There are some specific points that should be addressed in a revision:
1. The authors describe the inverse association of versican VI levels with the number of BCAS1+ oligodendrocyte lineage cells in MS lesions. BCAS1+ is expressed at high levels by immature oligodendrocyte (iOLs) but is also expressed by OPCs, representing ~20% of the total*

BCAS1-expressing pool. This should be clarified as unless cell morphology was also considered, BCAS1+ cells in the MS tissue may well represent OPCs as well as iOLs.”

We thank the Reviewer for this comment. We have described BCAS1+ cells as originally noted by the published paper (Fard et al., 2017 – reference 28). As concluded by Fard et al., BCAS1+ cells in the CNS segregate from OPCs and mature oligodendrocytes even though some BCAS1+ cells still express markers such as CC1 and NG2. To clarify this, we have now added the following text to the results on page 5 (lines 112-115):

“BCAS1+ oligodendrocytes have been described as newly formed, premyelinating oligodendrocytes that are distinct from OPCs and mature oligodendrocytes, even though the majority of BCAS1+ cells (76%) express a mature oligodendrocyte marker (CC1) and 16% of them are positive for an OPC marker (NG2)²⁸”.

“2. There appears to be no methodological description of the source and preparation of human OPCs described in Figure 1. The authors provide quantitative data on human OPCs and their response to culture on versican substrate. In addition to the number of O4+ cells, it would be useful to know the proportion of cells which express O4. To help characterize the human OPC culture in general, the proportion of Olig2-expressing oligodendrocyte lineage cells in these cultures should also be reported.”

We apologize for omitting the description of the human OPC preparations. This information is now added to the method section as a whole new paragraph on page 35.

It is an excellent suggestion to use the proportion of O4+ cells as additional readouts. The percentage of O4+ MBP+ cells is now shown in the newly added Figure 2 (panels H-L), as well as highlighted in the text on page 6 (lines 150-152).

To characterize the OPC cultures, cells were labeled for Olig2 and O4 markers. The proportion of Olig2-expressing oligodendrocyte lineage cells is now reported in the newly added Supplementary Fig. 3A and in methods on page 35 (line 1012).

“3. The authors note difficulty with low sensitivity of the spRNAseq. Perhaps, the use of RNAscope might be useful alternative technique to look at specific Tcell markers”.

We agree with the Reviewer that RNAscope provides better sensitivity and specificity to measure low-abundance RNA biomarkers. We have now mentioned it as one of our future experiments in the discussion section on page 15 (lines 428-435).

“4. Figure 2: While of very high quality, the demonstration of versican V1 expression in CD45+ cells do not seem to add much more conceptually than previously published in Brain. Having said that, the analysis of other ECM molecules such as versican V2 and aggrecan provide additional context and point toward the relatively specificity of versican V1 elevation in EAE.”

We thank the reviewer for pointing out the importance of studying different CSPG members. Our aim was to emphasize on the specificity of versican-V1 increase in lesion compared to other CSPG members. We hope that we have achieved this goal in the current manuscript.

“5. In Fig. 4, the EAE clinical data suggests a cohort specific difference between control and DIF groups prior to DIF treatment. It appears that animal group assignment was not appropriately randomized on the day of induction. While the effect of DIF in Fig 4 is relatively modest, the data presented in Fig 6 is far more convincing and somewhat alleviates these concerns.”

We appreciate this comment and the concern raised by the Reviewer. Despite our best efforts at randomization, mice in the 2 groups at onset of treatment in previous Figure 4 (now Figure 5, panel A) appeared slightly divergent, albeit not statistically different. We have now clarified this issue in the results on page 9 (lines 217-220) where we also immediately alerted readers of another experiment (Fig. 7) that follows. These sentences state the following:

“Despite our best efforts at randomizing mice into two groups at the beginning of their manifestation of clinical signs, mice appeared to be slightly divergent between the two groups although this was not statistically different (see Fig. 7 for another, longer term, experiment). This short-term treatment and prompt tissue harvest allowed us to study CNS T cell populations during peak clinical severity.”

“6. It is unclear how peak EAE was determined in each cohort and why DIF treatment was initiated at 12 days in Fig 4 and 13 days in Fig 6.”

On page 11 (line 279), where we first refer to ‘peak’ clinical severity, we now clarify that this is the period of pronounced clinical disability.

We apologize for the wrong labeling. In both experiments, treatment was started at day 12 (as shown by green box). The correction has been applied to Figure 7 (previous Fig. 6).

“7. Figure 5 shows that CSPG treatment of Th17 cells induces a more cytotoxic effect on OPCs. Can this effect be replicated by just incubation with versican V1? This would further support the premise that the relevant core protein is versican vs. another CSPG. Likewise, would versican V1-depletion prevent these effects in vitro? Additional experiments in this model could further define the relative importance of versican V1.”

This is an excellent comment. While there are no definitive means to selectively deplete versican-V1 from the CSPG mixture, we have now provided new data that versican-V1-exposed Th17 cells replicate the effects of CSPG-exposed Th17 cells in enhancing the killing of OPCs (new data for versican V1-exposed Th17 cells in Figure 6B,C). We also provide a new Figure 6D,E that the number of mature oligodendrocytes is reduced when OPCs are exposed to versican-V1-primed Th17 cells. We highlight these new results in the text (lines 249-251, 254-255).

“8. DIF and fluorosamine are fluorinated analogous of GlcNac and act by reducing the production of chondroitin sulfate side chains by inhibiting 4-epimerase. This compound will modulate both heparan sulfate and chondroitin sulfate GAG biosynthesis. Given that HSPGs have been recently implicated in contributing to the inhibitory ECM environment following demyelination (Saraswat et al., Nat Comm 2021), the interpretation that DIF acts solely via CSPG inhibition may not be justified in the absence of other data. Does DIF treatment alter HSPG sulfation, i.e. 10E4 staining? (i.e. Fig. 2I/P in the context of DIF treatment)”

We agree with the reviewer that DIF could affect both CS and HS GAG chain synthesis. In our previous paper (Stephenson et al., 2019), the effect of DIF on reducing HSPGs was modest. Here, we did not detect any significant changes as shown in newly added Supplementary Figure 8F,G. This is now highlighted in text (lines 330-332) on page 12: “The slight effect of difluorosamine on reducing the levels of HSPGs has been shown previously by our group; however, no change was found in the present study (Supp. Fig. 8F,G)”.

“9. Although assumed to be nuclei (DAPI?) there is no mention of NY (Fig 6. Panel G) in legend or methods.”

We apologize for this error. We have removed reference to nuclear yellow and substituted DAPI in its place.

“10. Figure 7. The use of the stub chondroitin-4-sulfate antibody is not described in the methods. Are all tissues treated with chondroitinase ABC prior to staining? If so, the reduction in versican VI core protein staining following DIF treatment suggests that appropriate GAG side changes alter the stability and/or proteolytic degradation of versican VI. Perhaps this could be discussed in the manuscript.”

We apologize for missing information. To detect the core protein of CSPG members (versican and aggrecan) and stub chondroitin-4-sulfate proteoglycan, sections were enzymatically digested with chondroitinase-ABC prior to staining. We have now pointed out this issue and described the antibody used for CSPG staining in the methods on page 30.

Minor:

“11. Introduction. Enhanced remyelination was also noted following chondroitinase treatment in Segel et al., Nature 2019.”

We thank the Reviewer for pointing this out. We have now discussed this finding by Segel et al. in introduction on page 3 (lines 67, 68): “The injection of the enzyme chondroitinase-ABC into the aged CNS promotes OPC proliferation and differentiation, and remyelination¹⁷.”

In response to Reviewer #2:

“Despite a considerable literature on changes in the extracellular matrix in MS, we still have a very poor understanding of how this matrix might enhance or perturb the parallel processes of damage and repair occurring in the disease. This paper examines one particular group of extracellular matrix components, the chondroitin sulphate proteoglycans and in particular versican. The paper starts with an interesting analysis of versican expression and oligodendrocyte numbers in MS lesions, showing an inverse correlation, and then moves onto experiments examining the effect of versican on TH 17 T cell polarisation/secretion and oligodendrocyte maturation. Then, using DIF to inhibit proteoglycan synthesis in a loss of functional approach the authors show a reduction in the extent of new oligodendrocyte formation in the EAE model of acute inflammatory demyelination.

The results are interpreted show an effect of versican both T cell polarisation and oligodendrocyte formation in vitro and in vivo. As such the paper represents a potentially valuable addition to the literature. However even if one accepts that the changes in T cell numbers are biologically significant (and see below) the key question is the relative contribution of the T cell changes and oligodendrocyte differentiation effects in the beneficial effects of DIF. This could surely be examined using the methodologies the authors have available; a comparison of the EAE and LPC models of demyelination following DIF administration. In the latter T cell polarisation is not involved in the generation of damage, and so comparisons would allow the contribution of the two possible effector mechanisms to be dissected apart.”

Thank you for these insightful comments. We appreciate the Reviewer’s suggestion to use LPC demyelination following DIF administration to exclude the contributions of T cells. Indeed, the data on the LPC model have been provided in a different paper by our group (Keough et al., 2016 – reference 16), showing the beneficial effect of a DIF analog, fluorosamine, on

remyelination. As T cells are not involved in LPC lesions, we proposed in that paper that blocking CSPG synthesis in the LPC model promoted remyelination through a direct effect on OPCs or by modulating monocyte/macrophages. Overall, given the various and complicated roles of CSPGs on multiple aspects of pathophysiology of EAE and LPC injury, we believe that direct comparison between these models would yield equivocal conclusions. Thus, in the current paper, we think it best to note that there are at least 2 cellular targets (T cells and OPCs) of CSPGs, rather than to dissect between the two. We hope the Reviewer agrees.

“In the experiments examining oligodendrocyte differentiation, versican reduces cell adhesion as the authors show. This would have significant secondary effects on morphological differentiation, making the latter experiments hard to interpret. Could the authors provide a uniformly adhesive substrate and assess the effect of added matrix in the culture medium?”

This is excellent advice indeed. To provide a uniformly adhesive substrate, we counteracted the negative charges of CSPGs by adding positively charged poly-arginine peptide, and the mixture was then adhered onto the bottom of the tissue culture well. This resulted in adhesion of OPCs onto the mixture comparable to control, but did not reverse the reduced process outgrowth. These new results are shown in Supp. Fig. 3B, C. Moreover, we increased the number of attached cells on CSPGs by using a very high initial seeding density; even with more cells attached, their process growth on CSPGs was still impeded (new Supp. Fig. 3D,E). These new data are also highlighted in the text in Results (page 6, lines 153-159).

We did not add matrix in a soluble form to the culture medium as the diffuse nature of the CSPG when diluted in the culture medium would not represent the local concentration that anchored cells would encounter. We hope the Reviewer agrees.

“The conclusion from the MS lesions that high areas of versican expression correlate with low numbers of new oligos requires a quantitative analysis (of ECM expression) rather than the qualitative approach taken in the manuscript”

We completely agree and appreciate this concern raised by the Reviewer. To avoid differences in background stain across different slides, we have normalized the versican expression in lesion to the normal appearing white matter (NAWM) of that tissue section. We have now described this at the end of the first paragraph of Methods (page 27).

“Figure 2D is confusing in that the dashed box on the left-hand panel doesn't correspond to the right-hand panel. Why does the versican labelling in the left-hand panel appear to be intracellular?”

We apologize for this. The Imaris 3D rendition distorted the original image so that the dashed box on the left-hand panel no longer lined up easily with the image to the right. To avoid misinterpretation, we have removed the dashed box in the left panel, and we display a different Imaris image on the right panel. In the figure legend (now Figure 3D), we now note that the panels are not meant to be of adjacent sections or same location, but rather to highlight the versican staining in the extracellular space and within CD45+ cells. As well, we now note in Results (line 172-175) that the intracellular versican staining may reflect cellular production, as we previously found versican-V1 transcript in CD45+ cells using in situ hybridization (Stephenson et al., 2018).

“In figure 3 the changes in the T cell populations are very modest, raising the question as to

whether this would have a significant biological effect. Equally in figure 4 the effect size in vivo is very small-about 4%. How do the authors conclude such small effects will have a major effect on myelin repair? The toxic effects of treated T cells in figure 5 is very interesting, but would the very small changes in population numbers seen in the earlier studies be sufficient, on the basis of data in figure 5, have an effect on oligodendrocyte numbers in vivo?"

These are great comments, and ones that we initially struggled with when we first obtained the results. However, the literature indicates that small changes in T cell numbers have pathologic significance. Despite the prominent role of T cells in MS pathogenesis¹⁻³, the recent single nucleus RNAseq study of the inflamed edge of chronic active MS lesions indicate that only 6% of immune cells are T cells⁴. Nonetheless, removal of the small proportion of double-positive Th17/1 cells from the CNS after treatment with natalizumab is associated with a dramatic decrease in the expansion of demyelinating lesions⁵. Besides, immune cells in the CNS of EAE mice comprise a relatively low proportion of Th17 cells⁶⁻⁸ (2-5%; in agreement with our study). Considering their low numbers but major contribution to EAE pathogenesis, it appears possible that any slight decrease in number of Th17 cells is adequate to ameliorate EAE and in the therapeutic settings⁹. We hope the Reviewer accepts our discussion.

1-Baecher-Allan C, Kaskow BJ, Weiner HL. Multiple Sclerosis: Mechanisms and Immunotherapy. *Neuron*. 2018 Feb 21;97(4):742-768. doi: 10.1016/j.neuron.2018.01.021. PMID: 29470968.

2-van Langelaar J, van der Vuurst de Vries RM, Janssen M, Wierenga-Wolf AF, Spilt IM, Siepman TA, et al. T helper 17.1 cells associate with multiple sclerosis disease activity: perspectives for early intervention. *Brain*. 2018 May 1;141(5):1334-1349. doi: 10.1093/brain/awy069. PMID: 29659729.

3-Moser T, Akgün K, Proschmann U, Sellner J, Ziemssen T. The role of TH17 cells in multiple sclerosis: Therapeutic implications. *Autoimmun Rev*. 2020 Oct;19(10):102647. doi: 10.1016/j.autrev.2020.102647. Epub 2020 Aug 13. PMID: 32801039.

4-Absinta M, Maric D, Gharagozloo M, Garton T, Smith MD, Jin J, Fitzgerald KC, Song A, Liu P, Lin JP, Wu T, Johnson KR, McGavern DB, Schafer DP, Calabresi PA, Reich DS. A lymphocyte-microglia-astrocyte axis in chronic active multiple sclerosis. *Nature*. 2021 Sep 8. doi: 10.1038/s41586-021-03892-7. Epub ahead of print. PMID: 34497421.

5-van Langelaar J, van der Vuurst de Vries RM, Janssen M, Wierenga-Wolf AF, Spilt IM, Siepman TA, Dankers W, Verjans GMGM, de Vries HE, Lubberts E, Hintzen RQ, van Luijn MM. T helper 17.1 cells associate with multiple sclerosis disease activity: perspectives for early intervention. *Brain*. 2018 May 1;141(5):1334-1349. doi: 10.1093/brain/awy069. PMID: 29659729.

6-Lee HG, Lee JU, Kim DH, Lim S, Kang I, Choi JM. Pathogenic function of bystander-activated memory-like CD4⁺ T cells in autoimmune encephalomyelitis. *Nat Commun*. 2019 Feb 12;10(1):709. doi: 10.1038/s41467-019-08482-w. PMID: 30755603; PMCID: PMC6372661.

7-Niedbala W, Besnard AG, Jiang HR, Alves-Filho JC, Fukada SY, Nascimento D, Mitani A, Pushparaj P, Alqahtani MH, Liew FY. Nitric oxide-induced regulatory T cells inhibit Th17 but not Th1 cell differentiation and function. *J Immunol*. 2013 Jul 1;191(1):164-70. doi: 10.4049/jimmunol.1202580. Epub 2013 May 29. PMID: 23720815; PMCID: PMC3785138.

8-Zhou L, Yao L, Zhang Q, Xie W, Wang X, Zhang H, Xu J, Lin Q, Li Q, Xuan Y, Ji L, Wang L, Wang W, Wang W, Shi T, Fang L, Zheng B, Li L, Liu S, Zhang B, Li X. REGγ controls Th17 cell differentiation and autoimmune inflammation by regulating dendritic cells. *Cell Mol Immunol*. 2020 Nov;17(11):1136-1147. doi: 10.1038/s41423-019-0287-0. Epub 2019 Sep 11. PMID: 31511643; PMCID: PMC7784850.

9-Lückel C, Picard F, Raifer H, Campos Carrascosa L, Guralnik A, Zhang Y, Klein M, Bittner S, Steffen F, Moos S, Marini F, Gloury R, Kurschus FC, Chao YY, Bertrams W, Sexl V, Schmeck B, Bonetti L, Grusdat M, Lohoff M, Zielinski CE, Zipp F, Kallies A, Brenner D, Berger M, Bopp T, Tackenberg B, Huber M. IL-17⁺ CD8⁺ T cell suppression by dimethyl fumarate associates with clinical response in multiple sclerosis. *Nat Commun*. 2019 Dec 16;10(1):5722. doi: 10.1038/s41467-019-13731-z. PMID: 31844089; PMCID: PMC6915776.

"In Figure 6 the affect of DIF on EAE severity is profound whilst in figure 4 it is marginal. Why are they so different?"

We appreciate the concern raised by the Reviewer. In Fig. 5 (Previous label: Fig. 4), treatment was short-term (5 days) while in Fig. 7 (Previous label: Fig. 6), we continued the drug for 15 days as we were focused on remyelination during the chronic phase. In both experiments, the mean clinical score decreased significantly after 5 days of treatment. The prompt tissue harvest after short-term treatment in Fig. 5 allowed us to study CNS T cell populations during peak clinical severity in control mice. We have now clarified this issue in the results on page 8 (lines 218-219) where we also immediately alerted readers of another experiment (Fig. 7) that follows. The sentence (line 220-221) now reads: “This short-term treatment and prompt tissue harvest allowed us to study CNS T cell populations during peak clinical severity.”

“Figure 7, which illustrates the key experiment of the paper, requires a quantitative analysis of cell numbers and re-myelination-the latter ideally by electron microscopy or by immunofluorescence to count the formation of new sheaths. May need more quantitation of the GFP”

We appreciate the reviewer’s suggestion for more quantitative analysis of cell numbers. We have now added new graphs in Fig. 8 (previous Fig. 7) showing new data of number of Olig2⁺ cells, Olig2⁺CC1⁺ cells and GFP⁺Olig2⁺CC1⁺ cells (new panels G-J).

We agree with the reviewer that EM is the gold standard for identifying thinner myelin sheaths as sites of remyelination in toxin-induced demyelination model. However, the method would only cover a small area of the spinal cord in a condition (EAE) where lesions could be widespread in the CNS, but occurring at uncertain locations. The advantage of using NG2^{CreER}:MAPT^{mGFP} mice is that the GFP signal informs on newly formed oligodendrocytes and myelin, and a large area of the spinal cord can be examined. We have now explained our rationale of not using EM in EAE in discussion on page 15, lines 406-414:

“Although the demonstration of thinner myelin sheaths by electron microscopy has become the gold standard for identifying sites of remyelination in localized toxin-induced demyelination models, it does not provide a ready means to study remyelination in EAE due to the uncertain location of lesions; thus, the microscale area of analysis in electron microscopy could easily miss a repairing lesion or under-report on remyelination across the spinal cord^{6,42}. Also, studies on myelin reformation in EAE have been hampered by the difficulty of assessing whether oligodendrocytes and myelin found in a lesion are spared or regenerated.”

“The conclusion at the end of the results that “particularly versican” reduce neuro inflammation and promoted remyelination is not justified, given that the key experiments used a broad inhibitor of proteoglycan synthesis”

We agree with the Reviewer, and we have now modified that conclusion by removing “particularly versican” from the sentence: “These results highlight for the first time that by lowering CSPG content within lesions, difluorosamine reduced Th17 neuroinflammation and promoted GFP+ profiles indicative of remyelination in EAE.”

In response to Reviewer #3:

“The current manuscript, “Versican impairs remyelination by inhibiting oligodendrocytes and promoting T helper 17 cytotoxic neuroinflammation”, aims to link several key observations about the inhibitory nature of CSPGs on remyelination. This area of work has potential impact for targeting and developing therapeutics for Multiple Sclerosis. This manuscript adds several points that were open questions from their previous studies of Keough et al. Nat Comm

publication in 2016 and Stephenson et al. 2018 in Brain. Here the authors further investigate the mechanisms underlying fluorosamine-mediated CSPG inhibition to promote myelin restoration. They extend their EAE studies by taking advantage of EAE combined with a recent strategy to label newly formed oligodendrocytes. This enables the authors to distinguish newly formed oligodendrocytes from existing oligodendrocytes. Importantly, this allows the authors the ability to assess immune cell-mediated demyelination as well as remyelination. Here the authors show the presence of a specific CSPG, VersicanV1, within lesions in patient samples, matching that to what is seen in EAE lesions. Further, the authors provide links to the T cell immune in response to Versican-V1 and CSPGs in vitro and CSPGs during remyelination.

The authors showcase a wide variety of complimentary in vitro and in vivo experimentation to examine the effects of CSPGs on T cell populations and EAE. I came away with the impression that the authors put in a lot of effort into this work. After reading the manuscript, I also had the impression that this is an extension of the authors previous work, Keough et al. 2016 Nat Comm and Stephenson et al. 2018 Brain, examining CSPGs and inhibiting CSPGs to promote remyelination. The new findings are interesting next steps to these previous publications. The key new findings 1) relate Versican-V1 levels and pre-myelinating cell densities in human post-mortem tissue; 2) demonstrate that Versican-V1 and CSPGs impact T cells; 3) demonstrate that CSPG synthesis inhibition reduces IL17/Th17 during EAE. Their previous work had demonstrated Versican-V1 levels correlate with EAE lesions and that CSPG inhibition reduced severity of EAE (which correlated with reduced Versican mRNA levels from EAE tissue samples), as well as enhancing new mature OLs in LPC remyelination models.”

Thank you for the remark that “the new findings of this paper are interesting next steps” to our previous publications. We appreciate the Reviewer’s support. We especially appreciate the Reviewer’s time in providing a very insightful and thorough review of our manuscript, the rebuttal of which has improved the content. We hope the Reviewer agrees.

“There are several major and minor points regarding the manuscript that should be addressed, however, by the authors. I outline these by topic, as follows:

Authors’ Conclusion1: versican-V1 is associated in lesions of MS with reduced remyelination One of the strengths to this work is the examination of post-mortem tissue samples from control and MS patients (Fig1A-E). As one of the key results, I would like to see the following clarifications to enhance the conclusions:

1. Have the example lesions been categorized? Could their categorization be labelled next to the images, please? It is unclear from the figure why certain regions were selected. Are you making conclusions about only active lesions?”

We appreciate the reviewer’s great suggestion. Lesion type and their corresponding expression of versican-V1 are now displayed as new Figure 1E. LFB and CD45 images of samples from Fig. 1A,B are presented in a newly added Supplementary Fig. 1, which clarify the selected lesion areas. We have now included more details in the results on page 5 (lines 102-110).

“2. On a related note, I am still unclear about whether Versican-V1 is primarily associated with demyelinating lesions, active vs chronic active lesions, or inactive lesions. Were the lesions only selected based upon LFB staining or were other classifications used (e.g., criteria that indicate active, inactive demyelinated, chronic active)? Do the Versican-V1 low vs high correlate with lesion type – inactive demyelinated, active demyelinating, chronic active, remyelinating, for example? If the authors have that information, that would also help paint the picture of how the

*Versican-V1 low vs high links (if at all) to lesion type. In Figure 1A, B the authors indicate looking at active lesions correlates with high V1, whereas the spRNASeq data indicates elevated levels of versican **with** inactive demyelinated lesions. Commenting further on this would help readers understand more clearly whether V1 levels may be relevant to demyelination and/or remyelination potential.”*

We thank the Reviewer for the insightful comments, and we apologize for not being clear. To improve clarity, first, patient information from Figure 1C-E is presented in the newly added Supp. Table 1. Second, to provide a better visualization of how versican-V1 levels and number of BCAS⁺ cells correlate with lesion type, samples in newly edited Figure 1E are color-coded by the type of lesions and labelled by their patient identifier number. Figure 1E should now clarify that the expression of versican -V1 is not dependent on the nature of lesions (i.e. active, chronic active and inactive lesions may all contain high versican-V1 immunoreactivity). However, in general, there is an inverse relationship between versican-V1 expression and BCAS⁺ cell numbers.

Accordingly, the last paragraph of page 5 now reads:

“Next, versican-V1 level of lesion was qualitatively divided into low versus high expression based on relative difference to versican-V1 immunoreactivity in NAWM in the small section. The comparison across 6 MS autopsy cases with 19 regions of interest (fields of view) shows that the expression of versican-V1 is not dependent on the type of lesion, with high or low versican-V1 being spread across active, chronic active or inactive white matter lesions (Fig. 1E). However, the number of BCAS1⁺ remyelinating oligodendrocytes is inversely related to versican-V1 content, with high versican-V1 areas generally containing a lower number of BCAS1+ cells (Fig. 1E).”

“3. It looks like MS-352 and MS-230 have high versicanV1 associated with CD45+ intensities, but MS-163 has high versicanV1 elsewhere in the brain section. Could the authors clarify these different observations?”

To clarify this matter, images of LFB and CD45/CD68 staining are presented in a newly added supplementary Figure 1, which verify the lesion areas. Versican-V1 immunoreactivity in MS-163 is not confined to the CD45+immune cells, as astutely noted by the Reviewer, and it expands outside the lesion. We believe that this is in line with our above (#2) rebuttal point that the expression of versican-V1 is not dependent on the type of lesion, and could thus be outside of lesion. This requires further definition in the future.

“4. Could the authors denote different t samples within their graph Fig1E (either by color-coding the dots or by different shapes)? That would help with the interpretation of how consistent these correlations were between different individuals (as opposed to different lesions). “

We thank the Reviewer for the great suggestion. The samples in the edited Figure 1E are now color-coded by the type of lesion and marked by their identifier label.

“Authors’ Conclusion2: V1 affects adhesion of OPCs and branched morphology of OLs The authors use isolated mouse OPC cultures and human OPCs to examine direct effects of Versican-V1 on the OL lineage cells. While this is an important experiment, the results leave the impression that Versican-V1 does not have strong effects directly on the OPC/OL cells. There are a few points I’d like the authors to address:

1. Can the authors comment on the discrepancy in protein concentration between mixed CSPGs and purified Versican-VI? I would predict that if Versican-VI was a main player in directly affecting OL biology, that Versican-VI would be more potent than the mixture of CSPGs. The results show the opposite, indicating to me that CSPGs collectively are much better at blocking OL branching rather than Versican-VI itself.”

This is an excellent question and one that we grappled with. We think that it is related to the mixed CSPGs containing a number of lectican CSPG members that collectively confer greater inhibitory potency. We have now discussed this in the Discussion section (page 14, lines 374-283). There, we have stated the following:

“We note that while the mixed CSPG preparation is very inhibitory for OPC process outgrowth at 2.5 µg/ml, much higher concentration of purified versican-V1 (20 µg/ml) is required (Fig. 2). A possible reason for this is that the mixed CSPGs contain a variety of lectican CSPGs (aggrecan, brevican, neurocan and versican, according to the manufacturer) which may act in concert to inhibit OPCs. Moreover, aggrecan has extensive glycosaminoglycan chains, far exceeding that of versican, and the additional negative charges should more effectively impair adhesion since the attachment of OPCs onto CSPGs is neutralized by the positively charged poly-arginine peptide (Supp. Fig. 3B, C).”

“2. What is the control being used? Previous work (Keough et al. 2015) showed that the decreased adhesion of OLs to CSPGs was due to the charge of the surface, not specific to CSPGs per se (Poly-arg peptide+CSPG coating neutralized the effects on OPC adhesion). For your Versican-VI experiment assessing adhesion (FigII), did you similarly control for non-specific effects of surface charge?”

To address the concern regarding the effect of negative charge, we performed an experiment using positively charged poly-arginine peptide to counteract the negative charges of CSPGs. The positive charged substrate improved adhesion of OPCs on CSPGs while mean process outgrowth remained unchanged. These results indicate the inhibitory role of CSPGs on process outgrowth regardless of the number of cells attached to surface. These results are shown in the newly added Supp. Fig.3B-E, as well as highlighted in text results on page 6 (lines 153-159).

“3. If pre-myelinating cells are reduced with Versican-VI in MS patient lesions, one would predict that VI inhibits OPC differentiation, prevents recruitment of new OLs, or does not support OL survival. Could the authors clarify how the in vitro experiment assessing branching of OL cells links to these predictions. An assessment of differentiation and/or cell survival would be more intuitive (not required but please comment/clarify).”

This is indeed an important point. While we have focused on morphological differentiation (process outgrowth) in culture, which is a necessary step in myelination in vivo given the need for the oligodendrocyte to extend elaborate processes that contact multiple axons to enwrap them, we have now performed new experiments at multiple time points to study maturation of OPCs through the acquisition of MBP. A new Figure 2 is thus presented. Our new data on MBP expression (Fig. 2H-L) show that versican-V1 and mixed CSPGs reduce the proportion of cells expressing MBP. We have now highlighted these findings in the text on page 6, lines 147-152.

“While process outgrowth in culture is indicative of the potential of an oligodendrocyte to extend elaborate processes to contact and enwrap multiple axons during myelination in vivo, the maturation of OPCs into myelin basic protein (MBP)⁺ cells is another important feature of remyelination. Thus, we addressed the proportion of O4-expressing cells that matured into MBP⁺

oligodendrocytes at 72h after plating, and this was decreased significantly on CSPGs or versican-V1 substrates (Fig. 2H-L).”

“4. While Versican-V1 at higher doses is quantified to affect outgrowth, the image in Fig1F is difficult to interpret. Could a higher magnification be shown of the cellular morphology? “

Thank you. Higher magnification images are now added to the bottom panels of new Figure 2A (previously Fig. 1F).

“5. The image in Fig1I with human cells looks like the cells are branched with Versican treatment. Is branching similarly affected for both mouse and human cells?”

Branching of adult human OPCs was not as noticeable as primary murine OPC culture and was not consistent among the different sample preparations. Accordingly, number of O4-expressing cells was reported instead of process outgrowth.

“Authors’ Conclusion3: Versican-V1 increases Th17 polarization

In contrast to effects on OPCs/OLs, the impact of Versican-V1 treatment and CSPG inhibition seems clearer for T cells (Figs3, 4). The specific role of versican-V1 vs other CSPGs was compared, which was great. Versican-V1 rather than other ECM found in the lesions (Fig2) enhanced Th17 production.

1. Could the author’s note the concentration of Versican-V1 and other recombinant proteins being used in these experiments to allow easier evaluation of the findings, please?”

The concentrations have now been added to the Figure legend. All were used at 10 µg/ml except for versican-V1 at 20.

“2. Could the authors clarify what is meant by “Data are presented as mean+/- SEM (n=3-4) and are from a single experiment representative of three independent experiments”? What is the “unit” for the “n” here (e.g., separate cell isolations from different mice)? Are data points shown from the three independent experiments or from one single “representative” experiment? Please expand the detail on this throughout the figure legends (also same statement in other legends).”

We apologize for the confusion that we created. Data points from tissue culture studies are from one single “representative” experiment and “n” is the number of replicates (wells) per condition in that experiment. The results were reproduced across three such experiments. Data from three different experiments are not pooled. We have now edited the figure legends to provide better clarity.

“3. While not necessary, it would have been nice to see the experiments in Fig3D-G done with purified Versican-V1, rather than mixed CSPGs. That would have made a more cohesive narrative for the novel conclusions regarding Versican-V1”.

We appreciate the Reviewer’s concern. Considering that versican-V1 is not commercially available, the limited amount of purified versican-V1 did not allow us to do these experiments. Nonetheless, we have prioritized the OPC-T cell co-culture experiment and used purified versican-V1 in a new experiment to confirm its cytotoxic effect that is similar to that for mixed CSPGs (Fig. 6B,C).

“Authors’ Conclusion4: cell polarization affect toxicity to OPCs

1. *Figure 5B: To my eye the number of lymphocytes between the column of Th17 and CSPG-exposed Th17 cells looks greater in the CSPG-exposed Th17 cell condition. Where the numbers of Th17 cells normalized between the conditions?"*

Thank you for bringing this to our attention. The number of T cells (2×10^4 cells per well) was the same in the different conditions. The previous images are now replaced as the revised Fig. 6B (previous Fig. 5B) includes new images from our recent experiments with versican-V1-exposed T cells.

"2. Although not necessary, again examining effects directly by Versican-V1 rather than CSPGs would have been even better."

We have repeated the experiments to include versican-V1. Our data show that higher proportion of OPCs incorporated PI when they were co-cultured with CSPG- or versican-V1-exposed Th17 cells for 12 hours. In another new experiment on OPCs that analysed cells at the end of incubation, we found reduced number of O4⁺ or O4⁺MBP⁺ cells in response to versican-V1-exposed Th17 cells (new Figure 6D,E).

"3. Was this experiment only conducted once with 4-6 wells per condition? I am confused by the legend "one biological replicate representative of three independent experiments". How is it representative of three independent experiments? Was each well a separate cell isolation from different animals?"

We apologize for the confusion. Data are from one "representative" experiment and "n" is the number of replicates (wells) per condition in that experiment. Experiments were repeated 3 separate times and data are not pooled. We have now changed the figure legend as below:

"n = 4-6 wells per condition in one experiment that was repeated three times. For each well, data from twelve images (FOVs) were averaged to a single data point."

"4. What is the relationship between Fig 5B and 5D (i.e., how was cell viability/toxicity actually measured)? Did the authors quantify the fraction of O4+ or O4+MBP+ cells that label with propidium iodide and/or loss of calcein AM? Was cell number per field of view the only measurable outcome in Fig5D? Please clarify."

We apologize for the confusion. The previous Figure 5B (now 6B,C) and 5D (now 6D,E) are different experiments where the intent of the former is the display of real time imaging data to show progressive killing over 12h, while the latter documents the number of oligodendroglial lineage cells remaining at 72h after exposure to Th17 or CSPG-exposed Th17 cells. The data of new Figures 6B,C and 6D,E are from new experiments that corroborate previous Figures 5B and 5D, but contain the additional group of versican-V1-exposed Th17 cells (and where the other groups were rerun in the new experiment). Yes, cell number is the only measurable outcome in new Figure 6E where for each well, data from twelve fields of views were averaged to a single data point. We have now clarified these in the manuscript (page 9,10) and in the legend to Figure 6.

"Author's Conclusion5: Difluorosamine improves EAE clinical score, altering immune cells and improving remyelination.

The demonstration of altering Th17 and IL17 with difluorosamine was an important component to this manuscript. While prior work from their lab showed improved EAE clinical score, this

goes the next step to identify changes that may underlie this and raise a new hypothesis about CSPG activation of Th17 cells during de/remyelination.

1. In Fig6: are the numbers of immune cells affected? How were the numbers assessed; some cell type markers are shown as % in WM or ROI and others as number per mm². It is therefore difficult to know exactly how this was assessed. Were statistics conducted to compare between control and DIF treatment. At a glance, it looks like %CD45⁺ in WM is increased in DIF treatment. Is it possible to assess CD4⁺ cell distribution in the EAE lesions, rather than solely by FACs in Figure4. That would give a better link between T cells and Th17 cells in relation to demyelination/remyelination.”

We thank the Reviewer for the great suggestion of assessing the frequency of CD4⁺ cells using IF staining. Representative images and graph for the number of CD4⁺ cells have been added to Fig.7H, K. The abundance of immune cell populations (CD45⁺, IBA-1⁺, CD3⁺ or CD4⁺) within the whole white matter of thoracic-spinal cord or within the lesion areas is now presented as cell number per mm² in Fig. 7I-K and newly added Supp. Fig. 8.

As pointed out by the Reviewer, there was an increasing trend of CD45⁺ cells. The concern regarding more monocyte/macrophages in DIF treated mice was addressed by looking at different immune subsets using RT-PCR, showing increased levels of markers for regulatory immune subsets.

“2. In Fig7: What time point is being assessed? It does not state it in the figure legend and/or figure itself. In DIF treatment, the appearance of CD45⁺ areas is reduced (Fig7A) in this example. This was used to define lesions. Therefore, I am unclear selected areas. How many mice were assessed for this experiment?”

We apologize for the lack of clarity. Figures 7 and 8 use samples from the same experiment; thus, the old Figure 7 (now Fig. 8) is of mice killed at day 26 of EAE. We have now explained this point in the figure legend. We have also noted in the legend to Figure 8 that the images of panel A are examples of slide scanner images. To enumerate the GFP⁺ areas, we have used two denominators. The first involved imaging the whole thoracic spinal cord section and then selecting the white matter region according to MBP staining (MBP staining of an adjacent section to define the white matter is not shown in figure); this enabled us to measure the percent area covered by GFP⁺ profiles in the white matter of 4 mice each in control and DiF group (Fig. 8C). The second denominator (Fig. 8D) was lesion in the thoracic spinal cord defined by CD45⁺ accumulation. Each lesion is a region of interest (ROI) and GFP expression was evaluated within each ROI. Here (Fig. 8D), there were 4 mice each in control and DiF group, with 8 – 10 ROIs per mouse. The figure legend has been rewritten to clarify these points.

“3. The images in Figure 7I are striking, with a clear difference in sheath-like extensions in the DIF treated example. The quantification was less clear; for the quantification in part J, what is meant by GFP⁺ particle size? How was the imaging and analysis done for Figure7? Please detail further how areas were selected and how ‘particle’ size was measured in the methods. Also, what are the n values used for the statistics; e.g., in J, is each dot representing a single imaging ‘field’ being treated as a ‘n’ for the statistics? This image is the best evidence given for remyelination, but quantification of remyelination has not been demonstrated. Instead, the current evidence demonstrates an increase in newly formed oligodendrocytes with processes. I would be hesitant to claim “higher content of new myelin” (line360) without demonstration of

node formation and/or electron microscopy of myelin sheaths. While I don't expect additional experiments per se, the text should be modified so that conclusions about myelin are not made."

We appreciate the Reviewer's comments. Particle size represents the average size of GFP⁺ profiles obtained through the ImageJ program (now described in an entire paragraph in Methods, page 31). To detail the results, the legend to Figure 8 is now substantially expanded. Descriptive sentences now include:

"C) Bar graphs (4 mice per group) comparing the extent of GFP⁺ in the white matter (WM), expressed as % of GFP immunoreactivity of the whole thoracic section; the WM was defined by MBP stain in an adjacent section (not shown). D) Bar graphs comparing the extent of GFP⁺ in region of interest (ROI), where each ROI is a lesion defined by area occupied by CD45⁺ cell. There were 4 mice per group, and 8 – 10 ROI per mouse. E) High magnification images of GFP⁺ cells and F) bar graph comparing average particle size of GFP in each CD45⁺ lesional ROI between DIF and vehicle treated mice (4 mice per group, and 8 – 10 ROI per mouse). H) Representative immunofluorescent images (z-stack) labeled for GFP (green), olig2 (white) and CC1 (red). G, I, J) Bar graphs comparing the number of oligodendrocyte lineage cells (olig2⁺), mature oligodendrocytes (olig2⁺CC1⁺) and GFP⁺ oligodendrocytes in the white matter as defined in panel C (4 mice per group). K) Representative immunofluorescent images (z-stack) labeled with GFP (green), versican-V1 (red) and stub chondroitin-4-sulfate attached to the core protein (gray). L, M) Quantification of the percent area occupied by immunoreactive CSPGs or versican-V1 in lesional CD45⁺ ROI (4 mice per group). N) Correlation analysis was performed between percent of GFP⁺ and CSPGs⁺ in the lesion ROI. Each dot represents one ROI from 4 mice per group."

We agree with the reviewer that the statement "higher content of new myelin" require more evidence, so we have deleted this part from our conclusion.

*"Overall Authors' Conclusion: "Our results identify versican-V1 as a therapeutic target in inflammatory demyelinating lesions to reduce Th17 polarization and its toxicity to OPCs, and to promote remyelination in the milieu of ongoing adaptive and innate neuroinflammation." My recommendation would be to **modify the title to reflect the direct findings from this work**. While all the evidence points to an inverse correlation of versican-V1 levels with pre-myelinating oligodendrocytes in lesions, and the ability for versican-V1 to enhance Th17 cell polarization, there are a few outstanding experiments to directly test the connection between these elements. The authors utilized many approaches that yielded results leading to the new hypothesis that versican VI promotes Th17 cytotoxic activity, which in turn decreases OL survival during remyelination. I would argue that this is currently a hypothesis supported by the data rather than a concrete conclusion. Independently, the experiments have demonstrated that 1) versican VI promotes differentiation of Th17 cells, 2) CSPGs promote Th17 IL-17 levels, 3) one effect of difluorosamine treatment in EAE is a decrease in Th17 IL-17 levels, 4) CSPG-exposed Th17 cells decrease OPC and OL cells in co-culture.*

There remain outstanding questions to directly test and bridge these conclusions, namely: whether Th17 cells are found in Versican-V1 lesions, whether blocking Th17/IL17 during remyelination would enhance myelin repair, whether decreasing Versican-V1 specifically (rather than CSPGs generally) would decrease Th17/IL17 during remyelination and correspond to an increase in OL number as well as increased myelin repair (rather than solely new OL formation). While I do not expect the authors to conduct these experiments at this time, these

would be considered important to really test the hypothesis. Therefore, it would be appropriate to modify the title and text to reflect this as a hypothesis drawn from the conclusions of the work rather than final conclusion itself.”

The Reviewer is very astute and has argued that several gaps remain to prevent us from being so forceful in our previous title of “Versican impairs remyelination by inhibiting oligodendrocytes and promoting T helper 17 cytotoxic neuroinflammation”. We agree with the Reviewer’s brilliant logic and concerns. Thus, we have softened the new title while retaining the important elements of putative remyelination, interference with oligodendrocytes, Th17 cells and cytotoxicity. The new title is:

“Versican as a potential inhibitor of remyelination: proposed mechanisms through impeding oligodendrocytes and promoting Th17 cytotoxic neuroinflammation”.

We were not able to detect Th17-related proteins or mRNAs in EAE or MS tissue by IF staining or spatial RNAseq methods, respectively because of technical issues. These points have been added to the revised manuscript (page 15, lines 428-435):

“The association between T cell subsets and levels of versican-V1 within lesions could be studied further using RNAscope that provides better sensitivity and specificity (relative to spatial RNA sequencing) to measure low-abundance RNA biomarkers such as IL-17 and ROR γ T. In addition to T cells, the changes in innate immune responses and how they impact remyelination have yet to be explored thoroughly in the context of versican biology. Our future studies using versican conditional knock out mice would help to clarify the specific effect of versican rather than total CSPGs on immune cells during remyelination..”

Regarding the Reviewer’s question of whether blocking Th17/IL17 during remyelination would enhance myelin repair, we have extended the discussion (page 14) and also cited a new study:

“Various T cell subsets impact remyelination differently. Regulatory T cells (Tregs) enhance remyelination whereas Th17 cells prevent myelin reformation^{22,36,37}. In vivo transfer of myelin-reactive Th17 cells reduces endogenous remyelination in a toxin-induced demyelination mouse model²². A recent study shows the direct contact between CNS-infiltrating Th17 cells and oligodendrocytes in EAE and MS lesions results in oligodendrocyte death and impaired remyelination through release of glutamate³⁸”

“Other points:

1. MAJOR POINT: The statements in the figure legends about n values, biological replicates, vs independent experiments need greater clarification. The addition of units would be extremely helpful. For example, instead of “n=3”, please state if “n=3 mice” or “n=3 lesions”. Also, it was unclear what was meant by phrasing such as “single experiments representative of three independent experiments”. This prevents appropriate evaluation of the results. Please give greater detail for each experiment shown.”

We have now done throughout the text and in the figure legend. Thank you.

Other examples in the figure legends:

a. Fig S2: there is currently no mention of the biological replicates/independent experiments conducted

b. Fig S3: n = 5 wells. Each well is a technical replicate rather than a biological/independent

experiment. If so (only technical replicate), statistical testing is not appropriate to draw a conclusion about the biology.”

We have now removed reference to statistics from Figure S3 (now Figure S5). As noted earlier, we have now added biological replicates and independent experiments to the figure legends.

“2. The authors have forgotten to include the methods regarding culture of human OPC/OLs.”

We apologize for the incomplete description of human OPC preparation. This information is added to the method section on page 35.

“3. This is stylistic, but I somewhat struggled with the flow/progression of the narrative starting from Versican-VI in MS tissue followed later by experiments that more broadly examined CSPGs in EAE immune response and remyelination. It made me wonder during all the later figures why Versican-VI wasn't the focus of experiments (instead of CSPGs broadly). My suggestion would be to write the manuscript in the order starting from the EAE model of remyelination with CSPGs – the effect on immune cells, then myelin, then focus in on T cells with CSPGs, how that narrowed down to Versican-VI affecting T cell Th17 populations, and Versican-VI on OPCs, then MS lesions with Versican-VI.”

Thank you for this advice. We did struggle with how best to describe this work. We had juggled the flow including in the manner that the Reviewer has advised. In the end, we settled with the current flow that begins with MS relevance, then the V1, and then finishing with the therapeutic strategy. We decided therefore to end with the difluorosamine results as we wish to leave the reader with thoughts of potential therapy. We are thankful that the Reviewer kindly mentions this as a stylistic issue, and hope that the Reviewer accepts our rationale.

“4. In the “Live imaging of OPC-Th17 cell co-culture”, 10,000 mOPCs are plated in a 96well, which is a reasonable cell density. However, under “Mouse oligodendrocyte precursor cells (OPC)” line 842, it states cells were plated at 100,000 cells per 96 well. That would be extremely dense. Is that an error?”

We apologize for this error, and the sentence has been corrected in the revised manuscript to reflect 10,000 cells per well.

“5. What does RFC stand for in Figure6? It may not be evident to your readers.”

It stands for “Relative Fold Change”. The abbreviation is now defined in the figure legend.

We thank the Reviewers for an outstanding job. We hope the Reviewers agree that the manuscript is now substantially improved, and ready for publication.

Reviewers' Comments:

Reviewer #1:

Remarks to the Author:

The authors have addressed all my concerns and points for further clarification.

Reviewer #2:

Remarks to the Author:

This manuscript is improved by attention to the reviewers and in particular I think that the change in title in response to the comments of reviewer 3 provides a much more balanced description.

In response to my comments the authors essentially accept that they cannot distinguish the relative contribution of altered TH17 positive cells and oligodendrocyte differentiation to the effects of versican. This is a pity, but I do accept the difficulty of comparing two models. However I remain concerned about the effect size of the TH17 depletion. If I've understood figure 5D correctly the depletion is from 15% to 12%. If this reduction in the TH17 population is thought to play a significant role in the versican effect, this is a remarkably small decrease to explain such an effect. The rebuttal seems to me to conflate two issues when the authors argue that the removal of the small percentage of TH17 positive cells by drugs can have a significant effect.

"removal of the small proportion of double-positive Th17/1 cells from the CNS after treatment with natalizumab is associated with a dramatic decrease in the expansion of demyelinating lesions⁵."

I'm sure this is right; but in the cited studies the majority of the (small number of) TH 17 positive cells are being removed, whilst the data in this study show only a very small % of such removal. I don't therefore think this supports the conclusion that

it appears possible that any slight decrease in number of Th17 cells is adequate to ameliorate EAE and in the therapeutic settings⁹. We hope the Reviewer accepts our discussion.

and I think this needs further clarification

Reviewer #3:

Remarks to the Author:

The authors clearly took reviewer comments on board, clarifying text and adding additional data to improve their manuscript. The additional quantification, details on MS tissue, additional purified versican-V1 in vitro experiments, and additional controls for cell adhesion of OPCs to CSPG-coated substrates have greatly strengthened the manuscript. I have one lingering point that I believe is easily addressed with data the authors have already obtained.

Final point:

The mechanistic insights rely on conclusions from the in vitro data to inform in vivo observations. This combination approach is very powerful. The authors have now clarified how they have displayed the data from in vitro experiments (thank you!): the data points on the graphs are technical replicates within a single representative experiment. With this clarification, in the interest of demonstrating the data robustness, the authors should show results between their replicates of the experiment (rather than simply replicate wells within a single experiment). The report of statistical analysis should be on experimental replicates rather than technical replicates (wells within an experiment would not be independent biological replicates). I applaud that the authors have technical replicates for the experiments, however, I would argue that individual repeats of the whole experiment represent the "n" for in vitro experiments. The authors state that the experiments were conducted three times each, so the data already exists. This should be a relatively quick modification/addition to the manuscript that would help assure readers of robust conclusions. This critical point should be simple to address using measurements already obtained by the authors (the hard part is done!).

On a similar thread, I would love to see/know that in vivo data in Fig 8 D, F. is analyzed per mouse for the statistics (rather than each ROI) as done in other in vivo data analysis in the manuscript.

I have enjoyed seeing how this manuscript has evolved and hope the authors take my final point on board to strengthen it.

Minor points on clarity:

To make it easier for readers, consider adding the identifier (e.g., MS-352; MS-230) above images in Fig1B.

Please check the alignment of the boxes in images that indicate magnified views
Several of these still seem off. For example: Fig 1B left; control in 2A.

Point by point rebuttal of NCOMMS-21-22835A, "Versican as a potential inhibitor of remyelination: proposed mechanisms through impeding oligodendrocytes and promoting Th17 cytotoxic neuroinflammation"

Reviewer #1 (Remarks to the Author):

"The authors have addressed all my concerns and points for further clarification."

Thank you.

Reviewer #2 (Remarks to the Author):

"This manuscript is improved by attention to the reviewers and in particular I think that the change in title in response to the comments of reviewer 3 provides a much more balanced description. In response to my comments the authors essentially accept that they cannot distinguish the relative contribution of altered TH17 positive cells and oligodendrocyte differentiation to the effects of versican. This is a pity, but I do accept the difficulty of comparing two models. However I remain concerned about the effect size of the TH17 depletion. If I've understood figure 5D correctly the depletion is from 15% to 12%. If this reduction in the TH17 population is thought to play a significant role in the versican effect, this is a remarkably small decrease to explain such an effect. The rebuttal seems to me to conflate two issues when the authors argue that the removal of the small percentage of TH17 positive cells by drugs can have a significant effect.

"removal of the small proportion of double-positive Th17/1 cells from the CNS after treatment with natalizumab is associated with a dramatic decrease in the expansion of demyelinating lesions⁵." I'm sure this is right; but in the cited studies the majority of the (small number of) TH 17 positive cells are being removed, whilst the data in this study show only a very small % of such removal. I don't therefore think this supports the conclusion that it appears possible that any slight decrease in number of Th17 cells is adequate to ameliorate EAE and in the therapeutic settings⁹. We hope the Reviewer accepts our discussion. and I think this needs further clarification."

These comments are very insightful. We agree that the reduction of the % of Th17 cells in the spinal cord caused by difluorosamine is small, but it is a consistent and statistically significant result. The small change may be related to our method of analysis where the entire spinal cord consisting of areas with or without lesion was removed for FACS analysis, and where around 10% of isolated cells in control mice were CD4+ cells (in difluorosamine: 6%). Of these CD4+ T cells, about 15% in control mice were IL-17 positive while DIF treated mice had around 12%. Thus, the number of IL-17 positive cells analysed by FACS is a small proportion of the initial cellular population, and its

denominator includes areas of the spinal cord that are not expected to contain lesions (e.g. gray matter). The measured reduction by difluorosamine (from 15% in controls to 12% in the drug group) is therefore likely to be an underestimation of their in vivo effect. We attempted to stain for IL-17 positive cells in the fixed spinal cord for immunofluorescence analysis, so that we would more accurately determine IL-17-positive cells that were only in lesional areas, but the staining was unreliable.

Thus, in the revised manuscript in the Discussion (second paragraph), we have added the following to discuss the reviewer's valid concern:

“We note that although statistically significant, the reduction of the % of Th17 cells in the spinal cord caused by difluorosamine (from 15% in controls to 12% in the drug group) is small (Fig. 5D). The minimal change could be related to our method of analysis where the entire spinal cord consisting of areas with or without lesion was removed for FACS analysis, and where around 10% of isolated cells in control mice were CD4+ cells (in difluorosamine: 6%). Of these CD4+ T cells, about 15% in control mice were IL-17 positive while DIF treated mice had around 12%. Thus, the number of IL-17 positive cells analysed by FACS is a small proportion of the initial cellular population, and its region of analysis includes areas of the spinal cord that are not expected to contain lesions (e.g. gray matter). The measured reduction by difluorosamine (from 15% in controls to 12% in the drug group) is therefore likely to be an underestimation. Attempts to stain for IL-17 positive cells in the fixed spinal cord for immunofluorescence analysis only within lesional areas were unsuccessful.”

Reviewer #3 (Remarks to the Author):

“The authors clearly took reviewer comments on board, clarifying text and adding additional data to improve their manuscript. The additional quantification, details on MS tissue, additional purified versican-VI in vitro experiments, and additional controls for cell adhesion of OPCs to CSPG-coated substrates have -greatly strengthened the manuscript.”

Thank you for improving our manuscript.

“I have one lingering point that I believe is easily addressed with data the authors have already obtained.

Final point:

The mechanistic insights rely on conclusions from the in vitro data to inform in vivo observations. This combination approach is very powerful. The authors have now clarified how they have displayed the data from in vitro experiments (thank you!): the data points on the graphs are technical replicates within a single representative experiment. With this clarification, in the interest of demonstrating the data robustness, the authors should show results between their replicates of the experiment (rather than simply replicate wells within a single experiment). The report of statistical analysis should be on experimental replicates rather than technical replicates (wells within an experiment would not be independent biological replicates). I applaud that the authors have

technical replicates for the experiments, however, I would argue that individual repeats of the whole experiment represent the “n” for in vitro experiments. The authors state that the experiments were conducted three times each, so the data already exists. This should be a relatively quick modification/addition to the manuscript that would help assure readers of robust conclusions. This critical point should be simple to address using measurements already obtained by the authors (the hard part is done!).”

We thank the reviewer for this suggestion. We have now reported fold change of all technical replicates over the three separate experiments. This has allowed us to pool the individual data from replicates across the experiments in Figures 2, 4 and 6; and has improved the robustness of data. We have changed the figure legend accordingly.

“On a similar thread, I would love to see/know that in vivo data in Fig 8 D, F, is analyzed per mouse for the statistics (rather than each ROI) as done in other in vivo data analysis in the manuscript.”

We thank the Reviewer for this comment. Figures 8D and F are aimed to show that the distribution of data throughout the studied ROIs. To address Reviewer’s concern, we have now added new graphs in Supp. Fig. 8H-I, showing data analyzed per mouse.

“I have enjoyed seeing how this manuscript has evolved and hope the authors take my final point on board to strengthen it.”

Thank you indeed for having improved our manuscript.

“Minor points on clarity:

To make it easier for readers, consider adding the identifier (e.g., MS-352; MS-230) above images in Fig1B.

Please check the alignment of the boxes in images that indicate magnified views

Several of these still seem off. For example: Fig 1B left; control in 2A.”

We thank the Reviewer for this accurate attention to details, and we have changed Figures 1 and 2 as suggested.

.

We truly appreciate all the constructive comments and suggestions from the Reviewers which have improved the quality of the manuscript. We hope the Reviewers agree that the manuscript is now ready for publication.

Reviewers' Comments:

Reviewer #2:

Remarks to the Author:

The addition to the discussion is very helpful - it clearly highlights the very small difference in th17 cells observed in the experimental group and provides a plausible explanation as to how this very small difference might underpin a biologically relevant effect. The reader can make her or his own mind up about the role of th17 cells based on this discussion, which is fine. So, I'm now happy to recommend publication of this interesting study

Reviewer #3:

Remarks to the Author:

The authors have addressed my comments.

Just a few suggestions regarding figure details to make it easier for readers to follow:

Image for MS-352 in Figure 1B is rotated between the merged (upper) image and the separate channel images (below). Please correct this.

Control image in Figure 2A – the lower, magnified image is not the area indicated by the box in the upper image. Please correct this. The other magnification boxes are out of alignment but at least are showing the same cells.

For Figure 4, could the authors label the y-axes to reflect the measurement (in addition to "fold change"), please?

I would recommend that the y-axes in Fig 5D should all start from 0 to avoid accentuating small changes, or at minimum show lines to indicate an axis break.

Rebuttal to Reviewers

NCOMMS-21-22835B, " Versican as a potential inhibitor of remyelination: proposed mechanisms through impeding oligodendrocytes and promoting Th17 cytotoxic neuroinflammation "

REVIEWERS' COMMENTS

Reviewer #2 (Remarks to the Author):

The addition to the discussion is very helpful - it clearly highlights the very small difference in th17 cells observed in the experimental group and provides a plausible explanation as to how this very small difference might underpin a biologically relevant effect. The reader can make her or his own mind up about the role of th17 cells based on this discussion, which is fine. So, Im now happy to recommend publication of this interesting study

We thank the Reviewer for the recommendation to publish.

Reviewer #3 (Remarks to the Author):

The authors have addressed my comments.

Just a few suggestions regarding figure details to make it easier for readers to follow:

We thank the Reviewer for agreeing that their concerns have been addressed.

Image for MS-352 in Figure1B is rotated between the merged (upper) image and the separate channel images (below). Please correct this.

We thank the reviewer for pointing out this. The correction has been applied to Figure 1B.

Control image in Figure 2A – the lower, magnified image is not the area indicated by the box in the upper image. Please correct this. The other magnification boxes are out of alignment but at least are showing the same cells.

We thank the Reviewer for this accurate suggestion, and we have replaced control images in Figure 2A.

For Figure 4, could the authors label the y-axes to reflect the measurement (in addition to “fold change”), please?

We have now added the complete information for the y-axes in Figure 4.

I would recommend that the y-axes in Fig5D should all start from 0 to avoid accentuating small changes, or at minimum show lines to indicate an axis break.

We thank the reviewer for this suggestion. The changes have been applied to Figure 5D.

We truly appreciate all the constructive comments and suggestions from the Reviewers which helped us to improve the quality of the manuscript. We hope the Reviewers agree that the manuscript is now substantially improved, and ready for publication.